# Applications of Tailored Mesoporous Silicate Nanomaterials in Regenerative Medicine and Theranostics

**DOI:** 10.3390/ijms26167918

**Published:** 2025-08-16

**Authors:** Jean Fotie

**Affiliations:** Department of Chemistry and Physics, Southeastern Louisiana University, SLU 10878, Hammond, LA 70402, USA; jean.fotie@southeastern.edu

**Keywords:** mesoporous silicates, bio-glass, bioengineering, smart nanomedicine, regenerative medicine, and theranostic applications

## Abstract

Tailored mesoporous silicate nanomaterials have attracted significant interest due to their exceptional surface properties, including high interfacial toughness, tunable thickness, customizable topology, optical transparency, and adjustable hydrophobicity. These characteristics enable them to exhibit a wide range of functional behaviors, such as antibacterial, anti-fouling, anti-fogging, lubricating, and abrasion-resistant properties, to name just a few. With recent advances in surface-modified nanosystems for bioengineering and biomedical applications, silica-based nanomaterials have emerged as promising candidates owing to their ease of surface functionalization, bioactivity, biocompatibility, biodegradability, and bioavailability. Consequently, they have been widely explored in various therapeutic contexts. This review provides a concise and concentrated summary of recent advances and applications of tailored mesoporous silicate nanomaterials in regenerative medicine and theranostics, with the primary focus being on how endogenous or exogenous triggers can be leveraged to achieve selective and precise delivery of various biomolecules and active therapeutics across diverse cellular environments, by harnessing the intrinsic properties of mesoporous silicate nanoparticles. This focus also guided the selection of specific examples provided to highlight their wide range of applications, with the report concluding with some perspectives and remaining challenges.

## 1. Introduction

Biomineralization is a natural process employed by both marine and terrestrial organisms to convert organic materials into inorganic composites. This transformation leads to the formation of biominerals such as calcium carbonates, calcium phosphates, and silicates, with oxides, hydroxides, and sulfates representing the most commonly observed crystalline and amorphous phases [1,2,3]. It is predominantly observed in marine species, and while its primary role is skeletal support, it also enables the development of sophisticated biological architectures that function as filters, grinders, or light-harvesters [1,2,3]. 

Biosilicification, a specific type of biomineralization, involves the transformation of dissolved silicic acid into solid silicates, leading to the formation of silica-based inorganic-biopolymer composites [1,2,3,4]. These structures, composed of amorphous or crystalline silicates, often exhibit remarkable mechanical properties like enhanced flexibility and fracture toughness, which, added to their intricate morphologies, may provide ecological or metabolic benefits to the organisms that produce them [1,2,3].

Consequently, extensive research has focused on replicating these inorganic–biopolymer composites, mainly through the synthesis of silica-based nanoparticles, as highlighted in recent review articles [4,5,6,7,8,9,10,11]. These nanoparticles are broadly classified into two types: nonporous and porous, with mesoporous silica nanoparticles (MSNs) being a particularly prominent and extensively studied subgroup within the porous category [4,5,6,7,8,9,10,11]. Mesoporous organosilicate nanoparticles are further divided into several families based on pore size and structural order. Among these are hollow mesoporous silica nanoparticles (HMSNs), characterized by a hollow core that offers a high loading capacity, organically modified silicates (ORMOSILs), whose surfaces are functionalized with organic molecules to enhance biocompatibility, periodic mesoporous organosilica nanoparticles (PMOs), featuring an ordered mesoporous structure with incorporated organic groups that expand their functional versatility, and dendritic silica nanoparticles (DSNs), which exhibit a branched, tree-like architecture that provides unique surface characteristics [4,5,12,13]. These structural variations enable this class of materials to exhibit a wide array of properties, including tunable pore size, surface functionalization, bioactivity, biocompatibility, biodegradability, and improved bioavailability [10,14,15,16,17,18,19].

The inorganic process of silicification is widely recognized to proceed via an SN2-like polycondensation reaction, wherein the high-energy, unstable monosilicic acid undergoes autopolycondensation, releasing water as a byproduct [4,20]. This stepwise mechanism leads to a sequential formation of dimers, trimers, and ultimately higher-order oligomers and polymeric structures (Q_0_ to Q_4_ species, as shown in Figure 1) [4,5,20]. During nucleation, these oligomers condense into thermodynamically stable silica nuclei—typically 1–2 nm in diameter—which then rapidly polymerize to form more complex silica architectures. Oligomers containing two or more monomer units frequently undergo cyclization, resulting in diverse structures such as cubic octamers and prismatic hexamers, with the broad range of Si-O-Si bond angles (135–160°) and bond lengths (1.58–1.62 Å) allowing for a broad range of structural motifs [4,5,6].

The pH of the solution also plays a crucial role in determining the higher-order morphology of the silicate, with near-neutral pH and low ionic strength conditions favoring the formation of a silica sol (dispersions of colloidal silica particles) [4,8,21]. Furthermore, numerous studies have shown that the choice of silicic acid precursor significantly affects the outcome of silicification processes [4,5,6,22]. Owing to the inherent instability of monosilicic acid, a range of alternative precursors, such as inorganic salts and organic acids, are frequently utilized to enable more controlled and reproducible silica formation [4,5,6,21,22,23,24,25,26].

Template-assisted sol–gel methods are widely employed for synthesizing these materials due to their ability to precisely control particle size, morphology, and pore architecture. This approach typically involves the hydrolysis and condensation of silica precursors to form Stöber-type silica nanoparticles, which frequently act as rigid templates [4,5,20,27,28]. To generate reactive silicic acids, alkoxide-based organosilica precursors must be acidified or alkalized, then polymerized into sol–gel networks that usually form stable, non-precipitating gel phases [4,5,20,26,27]. Common precursors such as tetramethyl orthosilicate (TMOS), tetraethyl orthosilicate (TEOS), and 3-aminopropyltriethoxysilane (APTES) often require the addition of co-solvents like methanol or ethanol to maintain homogeneity and prevent phase separation in aqueous systems [4,20,21,27,28]. Additionally, soft-templating strategies employ additives such as surfactants, micelles, vesicles, emulsions, or gas bubbles, comprising either hydrophilic or hydrophobic moieties, to guide the assembly of silica precursors into mesoporous structures through interactions with the soft template [5,8,16,29,30]. The template is later removed, typically via calcination or solvent extraction [15,16,28,31]. Figure 2 provides a comparative overview of differences in particle size, interfacial toughness, shell thickness, and surface topology of mesoporous silica nanoparticles, as influenced by the choice of precursor materials and synthesis conditions [32]. 

One of the key advantages of mesoporous silicate nanoparticles (MSNs) is the ease of surface modification, owing to the presence of reactive silanol (Si-OH) groups on their surface. These groups can be readily functionalized with a wide range of chemical moieties to enhance properties such as biocompatibility, stability, and targeting specificity [9,15,22,28,33,34]. Common surface modification strategies include the attachment of amine (-NH_2_) or carboxyl (-COOH) groups, polyethylene glycol (PEG), polysaccharides like chitosan, or polymeric coatings [8,13,15,25,28]. In addition to their low environmental impact and the affordability of starting materials, these various synthesis approaches enable the on-demand preparation of tailored materials with precise structural and functional properties, suitable for a broad spectrum of applications. 

As a result, mesoporous silicates have been widely utilized in smart drug delivery systems, primarily due to their ease of preparation, structural diversity, and the simplicity with which their surface can be functionalized or nanostructurally modified [7,13,35,36]. These characteristics allow for fine-tuning of pore sizes, making it highly adaptable for various delivery applications. The versatile applications of mesoporous silica for encapsulation, targeted drug loading, controlled release, and stimuli-responsive delivery are well documented in numerous recent reviews [5,13,15,28,37]. Notably, mesoporous silica has demonstrated effectiveness in a wide range of responsive delivery systems, including pH-responsive, thermo-responsive, glucose-responsive, ATP-responsive, magnetic-responsive, electro-responsive, and enzyme-responsive delivery [14,28,33]. More importantly, the designation of silica-based nanoparticles as “generally recognized as safe” by the U.S. Food and Drug Administration (FDA) has further accelerated their usage as diagnostic and therapeutic agents [28,37,38,39,40,41]. This review highlights recent advancements in the use of engineered mesoporous silicate nanomaterials in theranostics and smart regenerative medicine, specifically in osteogenesis, angiogenesis, and cementogenesis.

## 2. Application of Mesoporous Silicate in Regenerative Medicine

Unlike conventional treatments that primarily manage symptoms, regenerative medicine aims to restore or replace damaged cells, tissues, and organs caused by aging, disease, injury, or congenital conditions [42,43,44,45,46]. Its goal is to enhance the body’s natural ability to heal and repair itself [42,43,46,47]. This field holds the potential to treat, reverse, or even cure complex and currently untreatable conditions, such as cancer and organ failure, through advanced technologies like tissue engineering, cellular therapies, medical devices, and artificial organs [45,48,49,50]. This innovative approach is a highly interdisciplinary, merging knowledge from biology, chemistry, genetics, engineering, robotics, computer science, and medicine to create transformative healthcare solutions for some of humanity’s most pressing medical challenges [50,51,52,53]. 

As a result of their hydrophilicity, biodegradability, biocompatibility, porosity, and viscoelasticity, mesoporous silicate materials have found extensive applications in various regenerative processes, including osteogenesis (bone formation), angiogenesis (formation of new blood vessels, crucial for wound healing and tissue repair), and cementogenesis (formation of cementum, the mineralized layer covering tooth roots) [28,54,55]. These unique properties make mesoporous silicate highly suitable for a wide range of regenerative applications, including skin, cardiac (cardiomyocyte), muscle, cartilage, bone, and dental regeneration [28,54,55].

### 2.1. Application of Mesoporous Silicate in Hard-Tissue Regeneration

From a bone tissue engineering perspective, the design of complex biohybrids or biocomposites, as well as their surface modification, focus primarily on creating optimal conditions for skeletal support, ion exchange, mass transport, and the attachment of growth factors, drugs, and other osteogenesis-critical agents [56,57,58,59,60]. These enhancements establish an ideal substrate for the rapid nucleation and growth of calcium phosphates, as well as for promoting cell differentiation and proliferation [61,62,63].

In this context, John et al. [25,59,64] synthesized hexahedral, cage-like organic–inorganic siloxane core biohybrids featuring side chains fully functionalized with methacrylate derivatives, such as 2-hydroxyethyl methacrylate, ethylene glycol dimethacrylate, and 3-(trimethoxysilyl)propyl methacrylate. These materials demonstrated strong potential as 3D scaffolds for hard-tissue engineering, with the silicon-based systems providing a porous glassy (see Figure 3) conducive environment for biomineralization, promoting the formation of bone-like apatites and other calcium phosphates [25,59,64]. Additional studies, revealed that these hexahedral siloxane-based hybrids, with eight 3-(trimethoxysilyl)propyl methacrylate side arms, meet several critical criteria for hard-tissue applications, including appropriate chemical composition, dimensional structure, surface topography, and microstructural features [25,59,64]. In static in vitro bioactivity tests, a sustained release of silicon ions was observed from the polymer matrix. This release is considered advantageous, as silicon is an essential nutrient in bone metabolism [25,59,64]. Notably, silicon has been shown to reduce osteoclast activity while enhancing osteoblast proliferation, thereby supporting effective bone regeneration [61].

Bone defects resulting from traumatic injuries, surgical procedures, or congenital abnormalities represent the common clinical challenge that frequently necessitates hard-tissue regeneration [65,66]. Conventional bone graft substitutes, particularly those composed of calcium and phosphate, are widely regarded as ideal biomaterials and typically achieve satisfactory outcomes [65,67]. However, their regenerative effectiveness is often limited or unsuccessful in patients with underlying conditions such as osteoporosis or diabetes mellitus [68,69,70]. Additionally, bone reconstruction surgeries are at risk of complications from osteomyelitis—a bacterial infection that may necessitate systemic antibiotic treatment or even implant removal in severe cases [56,71]. As such, preventing bacterial infections remains a critical priority in orthopedic surgery, given the potential severe medical complications. Although rigorous hygienic protocols and prophylactic antibiotic treatments have significantly reduced the incidence of postoperative infections, a more sustainable and practical approach involves the localized delivery of antibiotics through orthopedic devices or prostheses made from synthetic materials with inherent antibacterial properties [56,62,63]. These materials can be loaded with antibiotics, antibacterial agents, or therapeutic metallic ions to enhance both their antimicrobial efficacy and biological performance [55,56]. From a broader perspective, various ions have been explored for specific therapeutic purposes: zirconium (Zr), boron (B), iron (Fe), magnesium (Mg), zinc (Zn), lithium (Li), strontium (Sr), cobalt (Co), copper (Cu), cerium (Ce), and gallium (Ga) have primarily been investigated for their role in osteogenesis [55,56,62,63]. For antibacterial activity, ions such as zinc (Zn), copper (Cu), cerium (Ce), gallium (Ga), and silver (Ag) are commonly used, while boron (B), cobalt (Co), and copper (Cu) ions have demonstrated angiogenic potential, supporting vascularization in bone tissue regeneration [54,55,56,62,63,72]. Figure 4 offers an overview of commonly used ions that support therapeutic functions, sanitation, and vascularization in hard and soft-tissue regeneration [56,73,74]. These localized drug delivery systems at the implant site offer promising advantages, including targeted delivery, sustained therapeutic effects, reduced systemic toxicity, and improved patient compliance [71,75,76]. Recent studies have also demonstrated that some injectable nanoparticles and nanocomposites possess intrinsic osteoinductive properties, capable of promoting the osteogenic differentiation of bone-forming cells [77,78,79,80].

Mesoporous bioactive glass nanoparticles (MBGNs)—a class of mesoporous silicates typically composed of silicon dioxide (SiO_2_), calcium oxide (CaO), and phosphorus pentoxide (P_2_O_5_) [81]—have gained significant attention in bone tissue engineering [62,75,82,83,84]. Their high surface area, excellent biocompatibility, and ability to modulate the release of bioactive agents make them particularly effective for bone regeneration applications [62,75,82,83,84]. MBGNs can also be engineered to support cell proliferation, differentiation, and new bone formation, and have shown efficacy in osteoporotic animal models [75,76,81,85]. Furthermore, the ordered mesoporous structure of these materials enables the encapsulation and co-delivery of therapeutic biomolecules and antibacterial metal ions, offering a synergistic approach to enhance antibacterial efficacy while promoting bone healing [57,62,86,87,88]. In a sense, this material has unlocked new possibilities for the intraosseous treatment of osteoporotic bone using minimally invasive surgery, with its main applications including (1) the use of bioactive glasses that modify the local physiological environment upon implantation, creating conditions unfavorable for bacterial growth, (2) doping the glasses with antibacterial metallic elements that are released during degradation to produce a bactericidal effect, and (3) incorporating antibiotics into the bioactive glass matrix to enhance antimicrobial efficacy [57,62,86,87,88].

Mesoporous silica aerogels have been widely used as carriers for the delivery of various antibacterial agents, including cinnamaldehyde, salicylic acid, and sorbic acid [89,90,91,92], many of which form uniform nanocrystals within the aerogel’s mesopores [92]. These systems have demonstrated strong antibacterial activity against a broad spectrum of bacteria, with effects lasting up to 45 days, highlighting their potential for long-term antibacterial performance [93,94,95]. For example, Lin et al. [96] developed a hierarchical scaffold by combining PEGylated poly(glycerol sebacate) (PEGS) with mesoporous bioactive glasses (MBGs) using a solvent-free urethane cross-linking via a spontaneous pore-forming approach. The resulting scaffold appeared to induce chondrogenic differentiation, enhance cartilage matrix secretion, and maintain chondrocyte phenotype. Furthermore, this material successfully repaired osteochondral defect by simultaneously promoting the regeneration of articular cartilage and subchondral bone, thereby restoring joint structural integrity and function [96]. 

Due to their ability to differentiate into various cell types, including bone, cartilage, and fat, bone marrow mesenchymal stromal cells (MSCs) are typically used in these experiments [57]. In fact, MBGNs loaded with the traditional Chinese medicine icariin were used as a long-term drug delivery system to promote the proliferation and osteogenic differentiation of bone marrow mesenchymal stromal cells (BMSCs), demonstrating potential for application in bone grafting procedures [57]. In another investigation, a composite scaffold comprising levofloxacin-loaded mesoporous silica microspheres made of nano-hydroxyapatite and polyurethane was designed to treat bone defects associated with chronic osteomyelitis [84]. Through immunohistochemical staining, PCR, and Western blot analysis, the scaffold was shown to exert both antibacterial effects and osteoinductive properties. This construct significantly enhanced the expression of osteogenic markers such as osteocalcin and collagen type I alpha 1 (COL1α1), and facilitated cell cycle progression from the G1 to G2 phase [84]. This composite scaffold also exhibited excellent biocompatibility in vitro, effectively promoting the proliferation and osteogenic differentiation of MSCs and MC3T3-E1 cells [84].

The biocidal properties of metal ions—particularly copper, silver, and zinc—commonly referred to as the oligodynamic effect, have long been employed in antibiotic-free mesoporous silica systems to inhibit microbial growth as part of long-term antimicrobial strategies [56,97]. In an in vivo study, Li et al. [93,98] evaluated silver-loaded multifunctional mesoporous silica nanoparticles and found that the material upregulated osteogenic-related factor expression. Notably, new bone formation was observed not only along the margins of the defect but also centrally, demonstrating the scaffold ‘s active role in bone regeneration [93]. Furthermore, the antibacterial efficacy of quaternary ammonium salt-modified core-shell mesoporous silica nanoparticles containing silver was investigated in vitro through co-culture with *Staphylococcus aureus*, *Escherichia coli*, and *Porphyromonas gingivalis* biofilms [99]. In this study, BMSCs were used to assess cytotoxicity, apoptosis, and osteogenic differentiation, and the results indicated that this nanosystem exhibits stable, concentration-dependent antimicrobial activity while maintaining biocompatibility [99,100]. 

In another study, Pérez et al. [62] compared the in vitro performance of mesoporous bioactive glass composed of 80% SiO_2_, 15% CaO, and 5% P_2_O_5_ with two analogous compositions containing 4% and 5% ZnO, both before and after loading with osteostatin—a bioactive peptide fragment (PTHrP 107–111) derived from parathyroid hormone-related protein, known for its role in bone metabolism and immunomodulation [62]. The Zn^2+^-doped MBG enhanced MC3T3-E1 cell viability and osteogenic differentiation to a greater extent than either osteostatin-loaded MBG or Zn^2+^-enriched MBG alone. These findings suggest that osteostatin synergistically amplifies the osteogenic effects of Zn^2+^, highlighting the potential of this combined approach in bone tissue engineering [62]. 

Micro-extrusion-based three-dimensional (3D) printing has also recently gained significant attention for the fabrication of composite nanomaterials, owing to its precise control over shape fidelity [101,102,103,104]. The incorporation of photo-crosslinkable gel components during the printing process enhances the mechanical stability of the resulting scaffolds [105,106]. Additionally, applying freeze-casting to the printed constructs further improves pore interconnectivity by generating hierarchically organized porosities within the scaffold struts—an essential feature for effective cell infiltration, nutrient transport, and tissue growth [105,107,108]. For illustration, a 3D-printed antibacterial, osteoinductive aerogel-based scaffold, produced through photo-crosslinking of methacrylated silk fibroin and methacrylated hollow mesoporous silica microcapsules, exhibited a significant potential in bone tissue engineering [108]. This composite system facilitated controlled antibacterial drug delivery, supported cellular ingrowth and proliferation, and promoted osteoblastic differentiation by upregulating osteogenic marker expression and enhancing matrix mineralization [108]. Similarly, Wang et al. [107,109] developed a 3D-bioprinted composite scaffold composed of MBG nanoparticles and polycaprolactone, loaded with doxycycline [107]. This scaffold exhibited dual functionality by simultaneously enhancing osteogenic activity and delivering broad-spectrum antibacterial effects [109].

For successful application in engineered constructs targeting large bone defects, biomaterials must facilitate the efficient exchange of oxygen, nutrients, and metabolic waste through newly formed vascular networks. To ensure and improve this requirement, a prevascularized modified hierarchical mesoporous bioactive glass scaffold was developed by seeding it with endothelial-induced adipose-derived mesenchymal stem cells (EI-ADSCs), in combination with osteogenically induced ADSCs, for the repair of critical-size bone defects [86]. To evaluate in vivo angiogenesis, green fluorescent protein (GFP)-labeled EI-ADSCs were used, with the labeled cells remaining viable seven days after subcutaneous implantation in nude mice, and contributed to neovascularization, as confirmed by double immunofluorescence staining for CD31 and GFP [86]. Further evaluation through dynamic bone formation analysis using sequential fluorescent labeling and Van Gieson ‘s picro-fuchsin staining, indicated that the prevascularized MBG scaffold exhibited the highest rate of mineral deposition [86]. This superior osteogenesis is likely attributed to rapid vascular anastomosis, which improved the survival and osteogenic activity of the seeded osteogenically induced ADSCs [86].

Mesoporous bioactive silicate nanoparticles have also garnered attention for their potential in promoting mineralization and enhancing the differentiation of human dental pulp stem cells (hDPSCs). For example, in a study evaluating the osseointegration of titanium-silica composite implants in a Göttingen mini-pig oral model, Katleen et al. [110] demonstrated that the infiltration of a SiO_2_ phase into titanium macropores allowed for controlled chlorhexidine release without impairing osseointegration. This functionalization approach appears promising for preventing and managing peri-implantitis [110]. In another study, silver-doped bioactive glass/mesoporous silica nanoparticles were evaluated for dentinal tubule occlusion, microtensile bond strength (MTBS), and antibacterial efficacy [111]. The composite exhibited acid-resistant tubule occlusion and maintained MTBS within a self-etch adhesive system [111]. Similarly, incorporating zinc-doped mesoporous silica nanoparticles into dental resin composites improved their mechanical properties and antibacterial performance [112]. 

Furthermore, curcumin-loaded mesoporous silica nanoparticles also demonstrated antibacterial efficacy against *Escherichia coli*, *Staphylococcus aureus*, and *Enterococcus faecalis* [113]. When used in implant fixtures, these nanoparticles significantly reduced bacterial colony counts, suggesting strong in vitro antimicrobial activity [113]. Similarly, in efforts to extend the lifespan of composite restorations by combating cariogenic biofilms and secondary caries, methacrylated mesoporous silica nanoparticles encapsulating chlorhexidine were developed [114,115]. These nanoparticles enabled controlled drug release and recharge without compromising mechanical properties such as flexural strength, surface roughness, and wear resistance. They also exhibited antibacterial activity against *Streptococcus mutans* and *Lactobacillus casei* [114,115]. In another study, mesoporous bioactive glass nanoparticles combined with graphene oxide (prepared via sol–gel and colloidal processing) demonstrated enhanced bioactivity and mechanical strength [82]. This composite upregulated key odontogenic markers (DSPP, DMP-1, ALP, MEPE, BMP-2, and RUNX-2) in hDPSCs, and activated the Wnt/β-catenin signaling pathway, indicating its potential to promote odontoblast-like differentiation and dentin formation [82].

Dentin hypersensitivity, often linked to exposed dentinal tubules, has also been targeted using graphene oxide quantum dot-coated mesoporous bioactive glass nanoparticles [116,117,118]. These spherical nanoparticles (~500 nm in diameter) achieved effective tubule sealing and rapid mineralization without inhibiting the release of Ca, Si, and P ions. Instead, the graphene oxide quantum dots facilitated hydroxyapatite formation, highlighting their promise for treating dentin hypersensitivity [116,117,118].

While mesoporous silicate materials offer significant potential in hard-tissue regeneration, owing to their high surface area, bioactivity, and controlled drug release capabilities as illustrated above, they are not without limitations. One major concern is their often-insufficient mechanical strength and structural stability, particularly for load-bearing applications [119]. As a result, they are frequently combined with polymers or integrated into composite systems to enhance mechanical performance [120,121]. Three-dimensionally printed scaffolds incorporating mesoporous calcium silicate have shown promising results, approaching the strength required for effective bone repair [107,109,122]. Another limitation lies in their degradation behavior. Although some materials degrade at a consistent rate, mesoporous silicates may not degrade in sync with new bone formation, potentially compromising tissue regeneration [10,123,124,125,126]. In addition, unfunctionalized mesoporous silicates have demonstrated some toxicity at high doses, likely due to the particles themselves rather than their degradation byproducts [125,127,128]. These materials can also elicit immune responses, particularly at high concentrations or when possessing positively charged surfaces, leading to inflammation that may further impede regeneration [7,16]. Furthermore, despite their well-documented advantages as drug delivery systems, mesoporous silicates carry a risk of premature therapeutic agent release, although this can be mitigated through surface modifications, controlled-release strategies, and the incorporation of stimuli-responsive mechanisms to ensure targeted and timely delivery [110,125,129,130].

### 2.2. Application of Mesoporous Silicate in Soft-Tissue Regeneration and Wound Healing

Wound healing is a natural physiological response to tissue injury, involving a complex and coordinated sequence of biological events. This process engages various cell types including cytokines, signaling mediators, and the vascular system to restore tissue integrity [131,132]. It begins with hemostasis, which is essential for controlling bleeding and initiating repair, followed by a closely monitored series of events including inflammation, cell migration, proliferation, extracellular matrix deposition, angiogenesis, and tissue remodeling [133,134,135]. While minor skin wounds typically heal within days, larger injuries, such as those from trauma, acute illness, or major surgery, can take weeks to resolve and often result in fibrotic scarring that may impair tissue functions [131,132,133,134,135,136]. Developing effective therapies to minimize scarring and enhance the healing of chronic wounds remains a critical and actively pursued area of research [106,137].

While various engineered materials have demonstrated effectiveness as wound dressings for infection control, many fall short in promoting the healing of chronic wounds due to insufficient blood vessel formation and limited cell proliferation, often resulting in prolonged inflammation [136,137,138]. Mesoporous silica nanoparticles (MSNs), known for their excellent biocompatibility and favorable surface properties, have garnered increasing interest for their superior hemostatic performance [139,140,141,142]. In fact, a multifunctional hydrogel scaffold was developed by incorporating methacrylated gelatin, methacrylated hyaluronic acid (HA), and MSNs, with gelatin providing mechanical strength and biodegradability, methacrylated HA enhancing biocompatibility and supporting cell adhesion, while MSNs function as sustained-release carriers, delivering bioactive compounds continuously to the wound site [139]. This hybrid hydrogel composite loaded with *Artemisia argyi* extract was proven to enable prolonged release of the extract, exhibiting strong antibacterial activity against common wound pathogens, while supporting the proliferation of human skin cells. Moreover, it promoted macrophage polarization toward the pro-regenerative M2 phenotype [139]. In vivo studies using a rat full-thickness wound model further confirmed the hydrogel’s therapeutic potential to accelerate wound closure, reduce granulation tissue formation, and enhance re-epithelialization, highlighting its effectiveness in promoting chronic wound healing [139].

Adipose-derived stem cells (ADSCs) have also been extensively investigated as a promising therapeutic strategy for rheumatoid arthritis (RA) due to their potent anti-inflammatory and immunomodulatory properties [143,144,145]. However, the direct use of stem cells poses concerns regarding immunogenicity and tumorigenicity [146]. Consequently, extensive efforts have focused on developing nanomedicine approaches utilizing ADSC-derived exosomes for cell-free regenerative therapies [146]. For instance, a delivery system was developed by physically adsorbing methotrexate into mesoporous silica nanoparticles, which were subsequently incorporated into ADSC-derived exosomes via ultrasonication [146]. This hybrid system was evaluated for its immunomodulatory effects, including its capacity to reverse macrophage polarization (from pro-inflammatory M1 to anti-inflammatory M2) and to inhibit the migration and invasion of fibroblast-like synoviocytes in vitro [146]. In vivo studies using adjuvant-induced arthritis and collagen-induced arthritis mouse models demonstrated targeted delivery to inflamed joints, sustained drug release, and significant reductions in joint swelling, synovial hyperplasia, and bone/cartilage degradation, highlighting the biocompatibility and rapid cellular uptake, and confirming its potential as an effective therapeutic platform for RA [146].

Spinal cord injury (SCI), another major cause of soft-tissue regeneration, continues to represent a major clinical challenge, primarily due to the limited intrinsic regenerative capacity of spinal neurons [147,148,149]. Reduced extracellular signal-regulated kinase (ERK) phosphorylation has been linked to poor axonal regeneration post-SCI. Interestingly, neurons exhibiting spontaneous axonal regrowth showed diminished expression of dual-specificity phosphatase 26 (DUSP26), a known negative regulator of ERK phosphorylation [147,148,149,150]. Harnessing this insight, researchers have developed a therapeutic system comprising DUSP26-specific inhibitors encapsulated in reactive oxygen species (ROS)-responsive nanoparticles, embedded within a photosensitive hydrogel matrix. This system effectively downregulated DUSP26 in primary neurons, leading to enhanced ERK phosphorylation and promoted axonal growth [150]. Upon transplantation into SCI mouse models, the platform enabled sustained drug delivery, targeted modulation of the DUSP26/ERK/ELK1 pathway, and facilitated axonal regeneration, with long-term effects including improved myelination, enhanced integration with host neural circuits, and significant recovery of motor functions [150].

Additionally, an innovative injectable bioactive hydrogel engineered using L-arginine-loaded mesoporous hollow cerium oxide nanospheres, has been proven to serve a dual function: modulating the oxidative microenvironment of SCI lesions by scavenging excessive ROS and continuously releasing nitric oxide, which promoted the neural differentiation of transplanted neural stem cells (NSCs) [151]. Mechanistically, this effect was associated with calcium influx-induced activation of the cAMP-PKA signaling pathway. The hydrogel also promoted microglial polarization toward the reparative M2 phenotype, supported the regeneration of myelinated axons, and enhanced the integration of transplanted NSCs with host tissue [151]. Ultimately, the system contributed to reduced neuroinflammation, inhibition of glial scar formation, and accelerated motor function recovery in SCI rat models [151].

Acute lung injury (ALI) is a critical inflammatory condition that can quickly escalate into acute respiratory distress syndrome, often resulting in irreversible lung tissue damage [152,153]. Effective therapeutic strategies must therefore focus on both anti-inflammatory and antioxidant interventions. In this context, a nanocomposite system featuring dual inflammatory targeting and immune evasion capabilities, consisting of epigallocatechin-3-gallate (EGCG) encapsulated within mercaptoketone-functionalized mesoporous silica nanoparticles effectively responded to elevated ROS by releasing EGCG in inflamed lung tissue [154]. In lung epithelial cells, the treatment also enhanced autophagy through activation of the MAPK/BNIP3 signaling pathway, and by simultaneously scavenging excess ROS and modulating autophagic activity. This targeted nanoplatform demonstrated significant potential for mitigating cellular injury and inflammation associated with ALI [154].

Erectile dysfunction (ED) is a prevalent male sexual disorder defined by the consistent inability to achieve or maintain an erection sufficient for satisfactory sexual performance. Among its various causes, cavernous nerve injury (CNI), particularly as a consequence of radical prostatectomy, represents a major source of iatrogenic ED [155,156,157]. The multifactorial nature of cavernous tissue damage in CNI-induced ED often leads to poor response and resistance to conventional vascular-targeted therapies [155,156,157]. A targeted nanotherapeutic system involving magnetic MSNs as drug carriers was loaded with self-assembling peptides RADA16-I and RAD-RGI to establish a neurotrophic microenvironment conducive to nerve regeneration [158]. Additionally, the neuro-targeting peptide HLNILSTLWKYR was grafted onto the surface of the MSNs to enhance selective delivery to injured nerve tissue. In vivo studies using a rat model of bilateral cavernous nerve injury demonstrated that this multifunctional nanocomposite significantly promoted cavernous nerve regeneration and effectively restored erectile function [158].

Silver nanoparticle-incorporated mesoporous silica granules (AgNP-MSG), fabricated via sol–gel processing, was also proven to be a highly blood-absorbent composite capable of inducing rapid hemostasis [159]. Structural analysis, cellular compatibility assessments, and adsorption studies demonstrated the material ‘s influence on in vivo degradability and hemostatic performance [159]. The composite exhibited excellent biocompatibility and sustained antibacterial activity, particularly against methicillin-resistant *S. aureus* [159]. In a rat liver injury model, 5% AgNP-MSG achieved effective hemorrhage control within 7 s, significantly faster than commercially available hemostatic gauze. Histological analysis confirmed complete degradation of the material at the injury site within two weeks [159]. In a separate study, an elastomeric scaffold developed using silver-doped mesoporous bioactive glasses incorporated into a poly(1,8-octanediol citrate) matrix via salt-leaching, demonstrated favorable biocompatibility with human dermal fibroblasts, indicating strong potential for applications in soft-tissue repair [88]. Another antibacterial platform involved silver-decorated mesoporous silica-coated single-walled carbon nanotubes, synthesized using an *N*-[3-(trimethoxysilyl)propyl]ethylenediamine-mediated method [160]. This nanoplatform exhibited potent antibacterial activity against multi-drug-resistant *E. coli* and *S. aureus* through bacterial membrane disruption and rapid silver ion release. In vivo studies using a rat skin infection model confirmed the material ‘s effectiveness in bacterial clearance, accelerated wound healing, and excellent biocompatibility [160].

In addition, a nanocomposite scaffold combining copper-containing MBGs with nanofibrillated cellulose (NFC) demonstrated high bioactivity in simulated body fluids and dose-dependent cytotoxicity on fibroblasts due to copper ion release [161]. This composite promoted angiogenesis in human umbilical vein endothelial cells (HUVECs), suggesting potential for chronic wound healing applications. The Cu^2+^ ions released from the aerogel significantly enhanced vascular formation in 3D spheroid HUVEC cultures and upregulated angiogenic gene expression in 3T3 fibroblasts, as confirmed by quantitative PCR analysis [161]. These findings underscore the scaffold’s proangiogenic capabilities when integrated into an NFC matrix [161].

Collectively, these examples highlight the diverse and promising applications of mesoporous silica-based nanomaterials in soft-tissue regeneration, wound healing, antibacterial therapies, and hemostasis. However, persistent concerns remain regarding biocompatibility, premature drug leakage, and potential toxicity, particularly associated with surface silanol groups. Additionally, low nuclease stability and the formation of protein coronas can negatively impact their therapeutic efficacy and biodistribution. As a result, researchers continue to work intensively to address these challenges and unlock the full potential of mesoporous silicates for soft-tissue regeneration.

## 3. Application of Mesoporous Silicates in Theranostics

In the era of personalized medicine, MSNs have emerged as a groundbreaking innovation in both therapy and diagnostics. Their ability to deliver drugs in a targeted manner significantly enhances therapeutic outcomes while reducing side effects on healthy tissues. A broad spectrum of therapeutic agents, including small molecules, proteins, DNA, RNA, genes, and antigens, have been successfully incorporated into engineered MSN-based systems. By leveraging the chemical versatility of MSNs, researchers have tailored their surface properties to improve drug loading capacity, enable controlled and targeted release, and enhance biocompatibility and bioavailability. These advancements have led to the development of smart nanocarriers that serve as multifunctional platforms for theranostics, integrating therapeutic and diagnostic capabilities (for specialized reviews, refer to [162,163,164,165,166,167]). As such, MSNs now hold promise for diverse biomedical applications, including drug delivery, bioimaging, and stimuli-responsive therapies, with particular emphasis on cancer treatment and personalized healthcare strategies. The following sections delve into specific applications of MSNs, with a primary focus on smart drug delivery systems and stimuli-responsive therapies, using recent reports to provide deeper insights into these various applications. Additional attention is given to their roles in bioimaging, photodynamic and photothermal therapies, cancer theranostics, gene delivery, and their expanding potential in the landscape of personalized medicine.

### 3.1. Application in Drug Delivery and Stimuli-Responsive Therapy

Conventional drug delivery methods often suffer from limitations such as poor pharmacochemical stability, uncontrolled drug release, and inadequate targeting, which collectively lead to low bioavailability, nonspecific distribution, reduced therapeutic efficacy, and increased risk of toxicity. Towards mitigating these limitations, advanced nanomaterials have been developed as innovative carriers to improve targeted therapeutic delivery and minimize systemic side effects [162,165]. As a result of their unique features, including high specific surface area, tunable pore sizes, robust mechanical properties, excellent biocompatibility, and the presence of abundant surface hydroxyl groups that facilitate easy functionalization, mesoporous silicates have garnered significant interest, particularly for use in exogenous activated therapies, and stimuli-responsive drug delivery systems, enabling site-specific, on-demand drug release in response to internal or external triggers [162,163,164,165,166,167].

Elevated intraocular pressure is a major pathogenic factor in glaucoma and is often associated with irreversible vision loss. In a comparative study, the transcorneal permeability of hollow mesoporous organosilica nanocarriers, functionalized with manganese (III) tetrakis(1-methyl-4-pyridyl)porphyrin and further modified with thioether, biphenyl, or thioether/phenylene moieties, was evaluated in porcine eyes [168]. Fluorescence intensity measurements in the aqueous humor revealed significantly enhanced corneal permeability according to the calculated diffusion, with apparent permeability coefficients suggesting that these modified nanocarriers could improve therapeutic efficacy in the treatment of primary open-angle glaucoma [168].

Etoricoxib, a nonsteroidal anti-inflammatory drug, suffers from poor aqueous solubility and is associated with gastrointestinal side effects such as bleeding and ulcers when administered orally [169]. To enhance its therapeutic profile, this drug was encapsulated in mesoporous silica nanoparticles via precipitation and solvent evaporation techniques. The resulting nanoparticles were incorporated into a carbopol–chitosan-based gel for transdermal delivery. In vitro, ex vivo, and in vivo evaluations demonstrated that the optimized nanocomposite gel provided a more sustained and higher release of etoricoxib compared to conventional etoricoxib gel [169]. Notably, it produced a significantly enhanced anti-inflammatory response, with marked reductions in IL-1β and TNF-α levels, with histopathological analysis confirming the absence of skin toxicity in treated subjects [169].

In another study, multichiral mesoporous silica nanoscrews (MCNSs) modified with L/D-alanine (L/D-Ala) via chiral templating and post-functionalization strategies were explored as oral drug delivery vehicles [170]. These chiral nanocarriers exhibited distinctive screw-like topologies with small cross-sectional areas, rough external surfaces, and excellent mucosal adhesion and mucus penetration capabilities. Upon encapsulation of racemic flurbiprofen, the chiral microenvironments created by L/D-Ala-modified MCNSs enabled enantioselective drug absorption in vivo, with the mesoscopic chiral nanochannels allowing for a controlled loading and release kinetics, demonstrating both structural and functional advantages over non-chiral counterparts [170].

Traditional thrombolytic therapies are often limited by risks such as uncontrollable bleeding and secondary vascular re-embolism [171]. To address these issues, a thrombin-responsive, sequentially targeted nanoplatform was developed using mesoporous silica nanoparticles modified with fucoidan via a peptide linker and loaded with urokinase and tirofiban [171]. The design leveraged an attack-defense-protection strategy: fucoidan enabled P-selectin-mediated targeting to thrombus sites, where thrombin-triggered fucoidan shedding led to rapid urokinase release for effective clot dissolution. Simultaneously, tirofiban was released gradually to prevent re-embolization [171]. In vitro and in vivo data confirmed reduced systemic bleeding, offering a promising approach for safer and more effective thrombolysis [171].

The clinical administration of carvedilol—a beta-blocker used off-label in pediatric cardiovascular therapy—is often limited by poor oral bioavailability due to pH-dependent solubility and extensive hepatic metabolism. In order to develop a transdermal permeability administration method, two types of mesoporous silica nanoparticles—MCM-41 and SBA-15—were used to load carvedilol at varying saturation levels to enhance its solubility [172]. At high saturation, carvedilol-loaded SBA-15 showed significantly improved dissolution, attributed to its amorphous state and large pore diameter. Ex vivo skin permeation studies further demonstrated that SBA-15 formulations substantially enhanced drug release, increasing transdermal flux by 62% over controls [172]. This suggests that carvedilol-loaded SBA-15 gel offers a viable pediatric-friendly transdermal alternative, bypassing hepatic metabolism and the bitter taste, while improving overall bioavailability [172].

These selected examples demonstrate the remarkable enhancement in pharmacokinetics and therapeutic efficacy achieved through the smart delivery of drugs using mesoporous nanosystems across various disease treatments. However, among all medical applications, cancer therapy has benefited the most from these advanced materials, particularly given the pervasive side effects associated with conventional cancer treatments.

Biomimetic nanomaterials consisting of hollow mesoporous silica nanoparticles coated with 4T1 tumor cell membranes designed to enhance the water solubility of docetaxel (a hydrophobic taxane chemotherapeutic) and to create a nanoplatform with homologous targeting and immune evasion capabilities, was evaluated in the treatment of breast cancer [173]. The adhesion glycoproteins of the membrane coating enabled specific binding to homologous cancer cells, facilitating targeted drug delivery to tumor tissues. Additionally, the presence of CD47 protein on the membrane surface allowed the nanoparticles to evade immune detection, with this biomimetic nanomedicine demonstrating enhanced tumor accumulation, reduced systemic toxicity, and improved therapeutic efficacy [173].

Similarly, to enhance the delivery of thymoquinone to MCF-7 breast cancer cells, mesitylene-based mesoporous silica nanoparticles were coated with phosphatidylserine-cholesterol liposomes, forming a system designed for controlled drug release [174]. The resulting sol–gel formulation exhibited a slow and sustained release profile, with cellular uptake studies using fluorescein isothiocyanate-labeled nanoparticles showing approximately a fivefold increase in uptake compared to the free drug [174]. Cytotoxicity assays demonstrated a marked decrease in cell viability, with significantly higher apoptosis rates observed in cells treated with the liposome-formulated thymoquinone compared to those treated with the free drug or control at both 24 and 48 h [174]. Zhang et al. [175] also designed a strategy to enhance the internalization of mesoposrous silica nanoparticles by functionalizing their surface with a transferrin receptor (TfR) aptamer, exploiting the overexpression of this short single-stranded sequences of nucleic acids on many cancer cell membranes. Cellular fluorescence imaging and flow cytometry analyses confirmed that TfR aptamer-functionalized MSNs achieved markedly higher uptake across three cancer cell lines (MCF-7, A549, and HeLa), demonstrating both improved efficiency and selectivity [175]. Moreover, these aptamer-modified MSNs exhibited significantly enhanced drug delivery capabilities compared to both unmodified and randomly modified MSNs at equivalent doses and incubation times, suggesting their promise for targeted delivery to TfR-positive cancer cells and improved therapeutic efficacy [175].

In another study by Sah et al. [176], mesoporous silica nanoscaffolds synthesized using a heat-assisted hydrolysis technique, and functionalized with poly(lactic-co-glycolic acid), were loaded with nanosized gefitinib. To further enhance the surface properties and biocompatibility of the nanocarriers, 1-oleoyl-2-hydroxy-sn-glycero-3-phosphocholine was incorporated. In vitro drug release and cytotoxicity analyses, along with in vivo biodistribution studies in albino Wistar rats, revealed that the gefitinib-loaded nanoscaffolds exhibited superior antitumor activity compared to the free drug [176]. The system demonstrated improved cellular uptake, enhanced biodistribution, and greater tumor suppression with reduced toxicity. Notably, a higher concentration of gefitinib was detected in the liver, suggesting efficient systemic distribution [176].

Nitric oxide is as a promising alternative to conventional anticancer agents due to its multifunctional therapeutic properties and low risk of inducing drug resistance. However, its high reactivity necessitates the use of macromolecular donors for controlled and targeted delivery. As a result, Grayton et al. [177] designed mesoporous silica nanoparticles coated with hyaluronic acid to enable both passive and active targeted delivery of nitric acid to cancer cells, leveraging the enhanced permeability and retention effect, and the specific interaction between hyaluronic acid and CD44 receptors, which are overexpressed in many tumors [178]. Compared to uncoated mesoporous silica nanoparticles, the hyaluronic acid-coated system exhibited faster nitric oxide release kinetics and lower overall payload capacity [177]. Furthermore, the hyaluronic acid-mediated targeting significantly enhanced intracellular nitric oxide delivery, increasing oxidative stress and triggering mitochondria-mediated apoptosis in melanoma cells. Cytotoxicity evaluations in human dermal fibroblasts indicated a high therapeutic index, with minimal off-target effects [177]. Confocal microscopy and fluorescence spectroscopy confirmed markedly enhanced internalization of hyaluronic acid-coated nanoparticles in melanoma cells relative to the uncoated counterparts [177].

The blood-brain barrier (BBB) is one of the most selective physiological gateways that tightly regulates the exchange of substances between the bloodstream and the central nervous system. While essential for maintaining brain homeostasis by restricting the entry of toxins and pathogens, this membrane also poses a significant challenge to the delivery of therapeutics for neurological diseases.

Mesoporous silica nanoparticles, due to their surface chemistry and biocompatibility, have emerged as promising candidates for the selective transports of bioactive molecules across the blood–brain barrier. For example, rhodamine B isothiocyanate-conjugated MSNs modified with short polyethylene glycol chains and varying positive surface charges provided by either polyethylenimine or trimethylammonium groups, were investigated using a chick chorioallantoic membrane model as an alternative to rodent models [179]. The results showed that trimethylammonium-modified MSNs, which exhibited a higher positive charge, were capable of efficiently crossing the BBB, while polyethylenimine-modified MSNs were less effective [179]. Although the weakly charged formulation demonstrated superior doxorubicin loading and a faster release profile, the strongly charged trimethylammonium-modified MSNs exhibited greater BBB permeability and drug transport efficiency. These findings were consistent in murine models, confirming their ability to traverse the BBB [179]. However, MSNs modified with long-chain polyethylene glycol showed reduced brain penetration, likely due to steric hindrance shielding the active trimethylammonium groups. Furthermore, using two-photon imaging, Chen et al. [180] revealed that the ligand-free PEGylated MSN (RMSN25-PEG-TA) nanoparticles remained in circulation for over 24 h, indicating significant systemic stability and extended vascular residence. When loaded with doxorubicin, the resulting formulation achieved a sixfold increase in brain accumulation compared to the free drug [180]. In vivo studies confirmed its ability to suppress orthotopic glioma growth and extend survival by over 28% in brain tumor models. Importantly, the formulation displayed an improved safety profile, with the LC-MS/MS proteomic analysis identifying a distinct protein corona on RMSN25-PEG-TA, with proteins such as apolipoprotein E and albumin potentially facilitating BBB penetration [180].

Beyond oncology, this nanoplatform has been explored for the treatment of Parkinson’s disease (PD), a neurodegenerative disorder characterized by oxidative stress and degeneration of dopaminergic neurons. Although levodopa remains the standard therapy, its limited bioavailability and lack of neuroprotective effects reduce its long-term efficacy [181,182]. To address these limitations, a dual-delivery system was developed by combining levodopa encapsulated in lactoferrin-modified organic-inorganic hybrid MSNs with curcumin loaded into a lipid bilayer shell, exploiting the synergistic neuroprotective potential of both agents [183]. In vitro studies using rotenone-damaged neuronal models demonstrated that the composite formulation significantly reduced oxidative stress, suppressed α-synuclein aggregation, and improved neuronal survival compared to monotherapies [183]. In vivo experiments in 1-methyl-4-phenyl-1,2,3,6-tetrahydropyridine (MPTP)-induced Parkinson’s disease mouse models confirmed superior brain targeting, enhanced motor function recovery, and minimal systemic toxicity, highlighting the system ‘s potential as a safe and effective therapeutic strategy for Parkinson’s disease [183].

In recent years, viruses—nanoscale entities primarily composed of proteins and nucleic acids—have garnered significant attention as promising nanocarriers for transporting therapeutic cargos across various physiological barriers. Their unique structural and functional features have inspired the development of novel nano-drug delivery systems [36,184,185,186]. For instance, adenoviruses and adeno-associated viruses have been extensively studied in clinical settings and are increasingly recognized as effective gene delivery platforms for a wide range of diseases [187,188]. Drawing inspiration from the remarkable infectivity and tissue-penetrating capabilities of viruses, researchers have engineered virus-mimicking mesoporous silica-based nanocarriers for precise, programmable drug delivery [7,30,32,189,190]. These viral-mimetic nanocomposites replicate both the architecture and functional characteristics of natural viruses. They exhibit virus-like interactions with both normal and cancerous cells and tissues, while their high surface area allows for enhanced drug loading capacity and superior cellular uptake efficiency [7,36,186,190]. 

In fact, Wang et al. [7] demonstrated that virus-like mesoporous silica nanoparticles with a spiky, tubular, and rough surface—synthesized through a single-micelle epitaxial growth method in a low-concentration surfactant oil/water biphase system—exhibited exceptional cellular uptake, rapidly invading living cells within minutes (<5 min) [7]. This enhanced internalization, along with prolonged blood circulation time, was attributed to the particles’ biomimetic morphology and unique internalization pathways. Notably, the epitaxial growth strategy employed in this study offers broad applicability for fabricating diverse virus-like mesoporous core–shell nanostructures, paving the way for their use in biomedical applications [7]. Figure 5a illustrates a cartoon-style comparison of the cellular internalization of three silica nanoparticle types—smooth nonporous, smooth mesoporous, and virus-like mesoporous—while Figure 5b shows confocal laser scanning microscopy images of HeLa cells following nanoparticles uptake after incubation [7].

In a related advancement, three types of dendritic mesoporous silica nanoparticles with similar particle sizes but varying pore sizes were engineered for degradation in simulated intestinal fluid and used to encapsulate Fenofibrate, a lipid-lowering medication [191]. These MSNs demonstrated significantly higher drug-loading efficiency compared to conventional excipients. Under accelerated storage conditions, their rigid mesoporous framework inhibited drug crystallization, overcoming aging issues commonly associated with traditional solid dispersions such as PVP K-30 and enhancing long-term stability [191]. Pharmacokinetic studies in rats revealed that the oral bioavailability of the MSN-based Fenofibrate capsules was 1.31 times higher than that of commercial Lipanthyl capsules [191]. Simvastatin, another widely used lipid-lowering agent with potent antioxidant and anti-inflammatory properties, is clinically limited by its poor water solubility and extensive metabolism by CYP3A4 enzymes [192]. In order to mitigate some of these limitations, this drug was encapsulated within 3D dendritic mesoporous silica nanoparticles (MSNs), enhancing its physicochemical properties and therapeutic efficacy [192]. The encapsulation led to the transformation of crystalline simvastatin into an amorphous form, which significantly improved its saturation solubility and enabled sustained drug release following a Fickian diffusion mechanism. In vivo studies in a poloxamer-407-induced hyperlipidemia model demonstrated that the formulation enhanced the drug’s antihyperlipidemic and antioxidant effects while providing substantial protection against oxidative damage [192]. Histological analyses of liver and aortic tissues showed nearly normal morphology, underscoring both the safety and therapeutic potential of the simvastatin-loaded nanoparticles [192].

Furthermore, the ability of mesoporous silica nanostars—with two distinct spike lengths—synthesized via a modified sol-gel single-micelle epitaxial growth method, to cross an in vitro blood-brain barrier multicellular model was evaluated and compared to that of conventional spherical nanoparticles [189]. These virus-like nanostructures demonstrated efficient BBB penetration and exhibited no cytotoxicity or immunogenicity in peripheral blood mononuclear cells and neuronal cells at concentrations up to 1 μg mL^−1^ [189]. Notably, nanostars with shorter spikes traversed the in vitro BBB model more rapidly than their longer-spiked counterparts and spherical particles, which primarily accumulated at the apical and basolateral sides, respectively [189]. Molecular dynamics simulations revealed that interactions with the nanostars increased the configurational flexibility of the lipid bilayer, facilitating membrane wrapping, while also suppressing natural membrane fluctuations [189]. These findings highlight the potential of virus-mimetic mesoporous silica nanostars as effective carriers for brain-targeted drug delivery and provide insights into their underlying mechanisms of interaction with lipid membranes [189].

Doxorubicin, an anthracycline antibiotic widely used in chemotherapy regimens for tumors such as breast cancer, leukemia, and lymphoma, exerts its effects by intercalating DNA and inhibiting topoisomerase II, thereby inducing apoptosis and halting tumor growth [193,194]. Yang et al. [190] demonstrated that virus-biomimetic mesoporous silica-based nanocarriers offer notable advantages over traditional MSNs. These nanocarriers, designed to mimic viral structure and function, displayed infectious virus-like properties toward tumor cells, including rapid and efficient cellular uptake within just 5 min—significantly faster than smooth-surfaced MSNs [190]. With their large surface area, they also showed high loading capacities for therapeutic agents. After loading with doxorubicin and surface modification with tumor-targeting Arg-Gly-Asp (RGD) peptides, the virus-like nanocarriers exhibited potent antitumor activity both in vitro and in vivo, significantly outperforming conventional MSNs [190]. All these findings underscore the potential of virus-inspired MSN designs as next-generation drug delivery systems for effective cancer therapy.

Stimuli-responsive nanoplatforms represent a class of smart drug delivery systems capable of responding to endogenous or exogenous triggers—such as pH, temperature, light, and magnetic fields—to enable localized and on-demand cargo release. These platforms hold significant promise for advancing personalized precision medicine. For instance, pH-responsive nanocarriers can selectively discharge therapeutic agents in acidic microenvironments commonly associated with infection or tumor sites [13,28,114]. Mesoporous silica nanoparticles (MSNs) have emerged as ideal candidates due to their high loading capacity and controllable release profiles, and as a result of their tailored framework composition, considering that recent studies in the field have moved beyond single-stimulus systems toward multi-responsive platforms, integrating multiple stimuli to enhance targeting precision and release control [13,28,114]. For example, Song et al. [142] engineered an MSN system functionalized with amino acids and loaded with rosmarinic acid, encased in a pectin coating. This design enabled dual-stimuli (pH and pectinase) responsive release, with kinetics fitting the Higuchi model, consistent with Fickian diffusion [142]. In another study, a superhydrophilic mesoporous nanomaterial synthesized using bismuth-oxoclusters, 2-methylimidazole ligands, and citrate, at high-temperature and under hypoxic conditions, resulted in a biocompatible and pH-sensitive carrier, which was effectively loaded with doxorubicin without requiring further surface modification [195]. In acidic environments mimicking tumor or endosomal pH, the platform disintegrated to release the drug, facilitating effective intravenous delivery while minimizing off-target toxicity [195]. Additionally, it provided excellent computed tomography imaging capabilities, achieving notable therapeutic outcomes in a mouse lung tumor model [195].

While exploring cellular internalization, cytotoxicity, and hemolysis performance of the fluorescent hybrid materials, Ge et al. [196] introduced a dual pH- and temperature-sensitive system by coating bimodal mesoporous silica cores with (2-(2-aminoethyl)-6-(dimethylamino)-1H-benzo[de]isoquinoline-1,3(2H)-dione)-doped poly[(*N*-isopropylacrylamide)-co-(acrylic acid)] as a fluorescent copolymer shell. The nanocomposite showed high ibuprofen loading and a controlled release of 83.1% at pH 7.4 and 25 °C, compared to 17.9% under strongly acidic conditions at 37 °C. Additionally, the nanocarrier demonstrated enhanced cellular uptake, favorable internalization, and low cytotoxicity, making it a promising multifunctional carrier [196]. 

Furthermore, the anticancer agent elesclomol was encapsulated within the open pores of magnetic mesoporous silica nanoparticles, which were subsequently sealed with gold gatekeepers, and the nanoparticle surface functionalized with polyethylene glycol and an epithelial cell adhesion molecule (EpCAM)-specific aptamer for improved biocompatibility and tumor targeting [197]. This platform exhibited a sustained, pH-responsive release profile over 72 h, with an initial burst release. In vitro studies demonstrated higher cytotoxicity against EpCAM-positive human prostate cancer cells (PC-3) compared to EpCAM-negative CHO cells, indicating selective targeting [197]. Furthermore, in a PC-3 xenograft mouse model, the targeted nanoparticles significantly inhibited tumor growth while reducing the systemic toxicity associated with free elesclomol [197].

Another system utilized mesoporous silica nanoparticles grafted with transferrin via pH-sensitive linkers to block pore openings at physiological pH and enable drug release in acidic tumor environments [198]. A complementary nanoplatform employed a pH-sensitive diimine bridge integrated into its framework, rendering the entire carrier degradable under acidic conditions. Doxorubicin release kinetics from both systems were evaluated at pH 7.4 and pH 5.0 using conventional models and a three-parameter model accounting for drug–matrix interactions [198]. Notably, the degradable system showed a shift from zero-order to first-order kinetics under acidic conditions, suggesting its potential for rapid drug release within cancer cells. Enhanced cytotoxicity was observed in A549 and H1299 lung cancer cell lines using the framework-degradable system compared to the transferrin-gated design [198].

Carboxymethylcellulose (CMC)-based mesoporous silica nanohydrogels developed by dispersing amine-functionalized MCM-41 (NH_2_-MCM-41) within a CMC matrix and crosslinking with Fe^3+^ ions, exhibited significantly greater swelling at pH values representative of intestinal and body fluids compared to gastric pH, confirming their pH-responsive nature [199]. Drug release from the CMC/NH_2_-MCM-41 system was slower than from the non-functionalized control, and cell viability assays confirmed their excellent biocompatibility for biomedical applications [199]. In an analogous study, in order to enhance the clinical potential of fucoxanthin—a hydrophobic compound with promising anticancer activity but low stability and bioavailability—a pH-sensitive delivery system was created using alginate oligosaccharide-encapsulated aminated mesoporous silica [200]. This nanoplatform demonstrated efficient drug loading and pH-responsive release, with 73.6% cumulative release at pH 5.5 after 24 h. Correspondingly, enhanced cytotoxicity was observed in HCT116 colon cancer cells, indicating improved therapeutic efficacy [200]. Similarly, a virus-like mesoporous silica nanoparticle platform designed to deliver repurposed metformin in a multi-stimuli-responsive, thermosensitive gel formulation for topical melanoma therapy, also exhibited sustained pH-dependent drug release—lasting 48 h at pH 7.4 and 6 h at pH 5.5 [201]. In vitro studies showed increased antiproliferative effects against A375 melanoma cells compared to free metformin. In vivo results demonstrated enhanced therapeutic efficacy, marked by elevated caspase-3 and NF-1 expression and decreased angiogenic markers such as VEGF and NRAS [201]. 

Self-propelling nanomachines based on Janus mesoporous silica nanoparticles have also been investigated as potential stimuli-responsive nanocarriers. One system, equipped with a pH-sensitive gate and powered by enzyme catalysis, used glucose oxidase immobilized via boronic acid–carbohydrate interactions [202]. In the presence of glucose, the enzyme catalyzed the generation of hydrogen peroxide and gluconic acid, which combined, triggered propulsion and pH-induced cargo release, successfully delivering doxorubicin to HeLa cells [202]. Additionally, a Janus nanocarrier comprising a mesoporous silica rod attached to a magnetic spinel ferric oxide core with an anatase titanium dioxide shell exhibited dual responsiveness to acid and microwave stimuli in another study [203]. Loaded with doxorubicin, it demonstrated excellent magnetic targeting, biocompatibility, and microwave-triggered drug release [203]. 

Starvation therapy targets cancer cell metabolism by cutting off critical energy sources. Beyond glycolysis, adipocyte-rich tumor microenvironments can promote fatty acid metabolism, which contributes to tumor growth and metastasis [204,205]. Targeting these metabolic pathways, a dual-starvation nanoreactor made of yolk-shell mesoporous organosilica nanoparticles loaded with a fatty acid transport inhibitor and conjugated with a breast cancer-targeting aptamer, was equipped with glucose oxidase catalyst, designed to disrupt glycolysis [206]. Once internalized by cancer cells, intracellular glutathione induced shell collapse and cargo release. Glucose oxidation generated gluconic acid and hydrogen peroxide, increasing intracellular acidity and oxidative stress [206]. Simultaneously, inhibition of carnitine palmitoyltransferase 1A by the fatty acid transport inhibitor suppressed lipid metabolism. This synergistic “dual-starvation” strategy effectively suppressed cancer proliferation while minimizing harm to healthy tissues [206]. Following the same path, metabolic reprogramming in cancer includes increased glucose uptake and lactate production even under normoxia—the so-called Warburg effect [207,208]. Similarly, a lactate-responsive nanoplatform created using Janus-type gold/mesoporous silica nanoparticles, was functionalized with lactate oxidase on the gold side and capped with α-cyclodextrin via arylboronate linkers on the silica side [209]. In the presence of lactate, lactate oxidase catalyzed hydrogen peroxide generation, triggering the cleavage of arylboronate linkers and uncapping the pores, resulting in a targeted drug release [209].

Diabetic retinopathy, affecting about 35% of diabetic patients, is linked to oxidative stress and inflammation, often exacerbated by vitamin D deficiency [210,211]. However, high doses of active vitamin D can lead to hypercalcemia. A glucose-sensitive delivery system using dextran-gated, 4-carboxyphenylboronic acid (CPBA)-functionalized MSNs was designed to release 1,25-dihydroxyvitamin D_3_ in a glucose-dependent manner [212]. The system leveraged the strong affinity of glucose’s 1,2-cis-diol groups to displace dextran from the CPBA moiety, thus uncapping the pores. Approximately 75% of vitamin D was released within 10 h under hyperglycemic conditions [212]. In high-glucose-treated retinal cells, this system enhanced vitamin D bioavailability and effectively reduced oxidative stress and inflammation, suggesting a strong therapeutic potential for diabetic retinopathy management [212].

Furthermore, hypoxia, characterized by low oxygen concentrations and the overexpression of reductase enzymes such as nitroreductases, is a hallmark of many solid tumors and contributes to their aggressiveness and resistance to conventional therapies [213,214,215]. As a result, a hypoxia-responsive MSNs-based hybrid nanoplatform loaded with doxorubicin, was functionalized with a nitroreductase-sensitive self-immolative gatekeeper [216]. Under bioreductive conditions—mediated by nitroreductase and NADH—the nitroaromatic gate undergoes reduction, triggering a self-elimination reaction that disintegrates the gatekeeper and releases the drug. In vitro studies in A549 cells, known for nitroreductase expression, confirmed the platform’s selective and efficient drug release under hypoxic conditions [216]. 

These selected examples illustrate how endogenous or exogenous triggers can be leveraged to achieve selective and precise delivery of various biomolecules and active therapeutics across diverse cellular environments, by harnessing the intrinsic properties of mesoporous silicate nanoparticles. In most cases, these strategies enhanced therapeutic efficacy while minimizing toxicity to healthy cells.

### 3.2. Application in Photodynamic and Photothermal Therapies, Bioimaging and Theranostics

A key component of selective and precise drug delivery strategies is the use of external stimuli, such as light or heat, as triggers. These exogenous cues not only enable controlled release of therapeutic agents but can also enhance their biological activity. Stimuli-responsive systems have been extensively explored in cancer treatment modalities, including sonodynamic, photodynamic, and photothermal therapies [217,218].

Chemodynamic therapy represents one such strategy, exploiting Fenton or Fenton-like reactions to generate reactive hydroxyl radicals within the tumor microenvironment. These radicals induce oxidative stress by disturbing the redox equilibrium, thereby inhibiting tumor cell proliferation and promoting cell death [219]. To improve the efficiency of CDT, a hollow mesoporous silica nanoparticle platform was designed, featuring a glutathione-responsive MnO_2_ shell [220]. At the tumor site, elevated glutathione levels trigger the degradation of the MnO_2_ coating, releasing Mn^2+^ ions. These ions subsequently react with the encapsulated dihydroartemisinin, initiating the Fenton reaction. This targeted nanosystem exhibited pronounced antitumor activity in vivo following subcutaneous administration in nude mice [220].

A nanodelivery platform combining hyperthermia and immunotherapy developed using gold nanorods coated with a silica shell and functionalized with folic acid for tumor-specific targeting also exhibited potent photothermal effects upon near-infrared irradiation, inducing apoptosis and suppressing tumor cell proliferation [221]. When loaded with the immune adjuvant imiquimod (R837), it further enhanced dendritic cell activation via hyperthermia, amplifying the immune response against cancer [221]. Alternatively, another multifunctional nanoplatform made of upconversion nanoparticles, was doped with gold combined dual photoluminescence and efficient photothermal conversion under near-infrared-I and near-infrared-II light [222]. Upon 808 nm excitation, this system exhibited strong luminescence, effective heat generation, and enabled high drug loading with controlled release, further highlighting the potential of mesoporous silica shells for near-infrared-triggered, image-guided photothermal therapy and precision drug delivery [222].

In another approach, multifunctional and biodegradable mesoporous silica–alumina nanoplatforms synthesized from rice husk-derived sodium silicate, effectively encapsulated doxorubicin and provided sustained drug release in simulated gastric and intestinal fluids [223]. Their strong optical absorption (250–420 nm) and broad near-infrared photoluminescence (680–900 nm) make them well-suited for fluorescence imaging. Importantly, the drug-free platforms were non-toxic to fibroblast and Caco-2 cells, while doxorubicin-loaded systems selectively inhibited up to 80% of Caco-2 colorectal cancer cells after 72 h [223]. Additionally, another near-infrared-responsive drug delivery system engineered using azobenzene-modified mesoporous silica-coated upconversion nanoparticles, was loaded with the chemotherapeutic agent [Ru(terpyridine)(dipyridophenazine)(H_2_O)]^2+^ (Ru(tpy)DPPZ) [224]. Upon 808 nm excitation, the upconversion nanoparticles emitted UV and blue light, triggering reversible cis–trans isomerization of azobenzene to enable controlled, on-demand drug release. Imaging studies confirmed the targeted delivery of both the nanoparticles and Ru(tpy)DPPZ to the nuclei of MCF-7 breast cancer cells, leading to DNA damage and efficient tumor cell destruction [224]. Given the nucleus’s central role in gene regulation and therapeutic response, these near-infrared-activated nanohybrids represent a promising platform for precision-targeted cancer theranostics [224]. 

While fluorescence imaging plays a critical role in tumor staging and therapy planning, indocyanine green still faces limitations in stability and targeting. Towards addressing this shortcoming, a nanocomposite probe was designed by conjugating anti-PD-L1 antibodies to PEG-coated mesoporous silica nanoparticles loaded with indocyanine green [225]. This platform exhibited strong and stable near-infrared-II fluorescence under 808 nm irradiation, improved photostability, biocompatibility, and prolonged tumor retention. Through both passive (EPR) and active (anti-PD-L1) targeting, the probe selectively accumulated in PD-L1-positive tumors in nude mice, achieving rapid, high-contrast tumor imaging with a tumor-to-muscle ratio of 3.94 [225].

Several other innovative mesoporous silica-based nanoplatforms have been implemented in the enhancement of cancer therapy through multifunctional integration. A biodegradable system combining chemo- and photothermal therapy constructed using organic mesoporous silica with tetrasulfide bonds, a copper sulfide core, and folic acid modification for targeted delivery, enabled controlled drug release, efficient tumor degradation, and strong photothermal conversion, resulting in potent in vitro and in vivo antitumor effects [226]. Furthermore, a mesoporous silica nanosystem coated with *Haliclona* sp. spicules designed to enhance transdermal delivery of protoporphyrin IX, led to deep skin penetration upon topical application, and completed primary melanoma eradication within 10 days in a photodynamic therapy of metastatic melanoma in mice, with no recurrence over 60 days [227]. In a similar study, a sequential transdermal delivery strategy based on a functional Deep Eutectic System (DES) with 2-deoxy-D-glucose as a hydrogen bond donor, and glutathione-responsive mesoporous organosilicon nanoparticles, disrupted tumor glycolysis, alleviated hypoxia and immunosuppression, and enhanced photodynamic therapy with MON@Ce6, yielding strong synergistic antitumor effects through immunogenic cell death [228].

Furthermore, a multifunctional platform devised around mesoporous silica nanoparticles coated with magnetic Gd-Zn-Cu-In-S/ZnS quantum dots, loaded with epirubicin, capped with gold nanoparticles, PEGylated, and functionalized with EpCAM aptamers demonstrated selective targeting, effective tumor accumulation, and enhanced therapeutic, with imaging capabilities both in vitro and in vivo [229]. In vitro studies demonstrated selective cytotoxicity toward HT-29 cancer cells over CHO cells, with significantly enhanced cell death under irradiation [229]. Additionally, to overcome radiotherapy limitations, another platform incorporating Mn/ZnO_2_ into mesoporous silica for targeted, pH-responsive delivery and Magnetic Resonance Imaging (MRI) enhancement, alleviated tumor hypoxia, normalized vasculature, and synergized with radiotherapy, leading to complete tumor regression and extended survival in mice [230].

Sonodynamic therapy (SDT) is a non-invasive cancer treatment that combines low-intensity ultrasound with a non-toxic sonosensitizer to selectively induce cancer cell death. A number of mesoporous silica-based nanoplatforms have been engineered to enhance SDT efficacy and tumor targeting. One such platform involves phthalocyanine-conjugated mesoporous silica nanoparticles designed to induce pyroptosis in liver tumors and boost anti-PD-L1 immunotherapy [231]. Surface modification with c(RGDyC)-PEG improved tumor targeting and systemic clearance, and when co-loaded with doxorubicin, the system synergized with PD-L1 blockade to significantly reduce both primary and metastatic tumor burdens [231]. Another multifunctional nanoplatform targeting hepatocellular carcinoma, co-delivered curcumin (as a sonosensitizer), doxorubicin, and perfluoropropane gas encapsulated within hollow mesoporous silica nanoparticles [232]. Biotin and acid-sensitive *cis*-aconitic anhydride-polyethylene glycol (CDM-PEG) coatings enhanced tumor targeting and pH-responsive drug release. Ultrasound exposure triggered ROS production via curcumin, and the nanoplatform inhibited drug resistance through P-glycoprotein suppression and improved imaging and therapeutic outcomes, all with high biocompatibility [232].

A thermosensitive mesoporous silica nanoparticle system based on astragaloside IV and 1-tetradecanol as a thermal gatekeeper, functionalized with PEG, PEI, and a tumor-targeting *Bifidobacterium bifidum*, enabled hypoxia-targeted delivery [233]. Focused ultrasound ablation surgery (FUAS) triggered localized drug release, enhancing tumor ablation through improved acoustic impedance and prolonged tumor site retention, with minimal systemic toxicity [233]. Additionally, a nucleolin-targeted hollow silica nanoplatform developed for lymphoma therapy, co-loaded with doxorubicin and the sonosensitizer indocyanine green, with the surface functionalized with PEGylated AS1411 aptamer–lipid conjugates, selectively targeted lymphoma cells via nucleolin recognition [234]. Under ultrasound irradiation, the system induced apoptosis through combined ROS generation and DNA damage, with strong accumulation at tumor sites in vivo via both the EPR effect and nucleolin-mediated uptake, and minimal off-target toxicity [234].

Additionally, mesoporous silica nanoparticles derived from silica-rich rice husk were also developed for cancer theranostics by incorporating europium and gadolinium, enabling dual-modal fluorescence and MRI. Surface functionalization with folic acid and AS1411 aptamer enhanced tumor targeting, while camptothecin loading provided therapeutic capability [235]. The system enabled glutathione-responsive intracellular drug release, resulting in selective uptake and increased cytotoxicity in HeLa cells compared to normal L929 fibroblasts [235].

Beyond oncology, this theranostic strategy has also been applied to enhance blood–brain barrier (BBB) penetration and enable sustained drug delivery for managing traumatic brain injury. A chalcone-functionalized mesoporous silica nanoparticle system incorporating diethylenetriamine pentaacetic acid as a chelator to target Aβ42 aggregates, exhibited strong interaction with Aβ42, as confirmed by a 30 nm red shift and a fourfold increase in fluorescence intensity [236]. The nanoparticles exhibited high biocompatibility, maintaining over 90% cell viability in PC12 (rat adrenal pheochromocytoma) and HEK-293 (human embryonic kidney) cells after 48 h. Furthermore, radiolabeling with ^99^ᵐTc achieved over 99% purity, and single-photon emission computed tomography imaging in rabbits confirmed successful brain penetration [236]. In a non-surgical repeated mild traumatic brain injury (rmTBI) mouse model with Aβ42 overexpression, histological analysis verified plaque formation, while biodistribution studies demonstrated enhanced brain uptake. Additionally, curcumin-loaded nanoparticles showed biphasic sustained release, underscoring the platform’s potential for targeted imaging and controlled drug delivery in neurodegenerative disorders [236]. 

This strategy has also been applied to the development of stimuli-responsive smart delivery systems for antibacterial agents. For instance, the photothermal efficiency of two-dimensional rhenium disulfide (ReS_2_) nanosheets was significantly enhanced when coated onto MSNs via an in situ hydrothermal reaction. This hybrid structure exhibited excellent light-responsive behavior and enabled controlled drug release [237]. Under laser irradiation, the system demonstrated approximately 99% inhibition efficiency against both Gram-negative (*E. coli*) and Gram-positive (*S. aureus*) bacteria. When loaded with tetracycline hydrochloride, a synergistic antibacterial effect was observed, achieving complete (100%) eradication of *E. coli* [237]. Similarly, mesoporous silica-supported silver-bismuth nanoparticles designed for synergistic antibacterial therapy showed outstanding efficacy against MRSA. The bismuth-induced hyperthermia facilitated cell membrane disruption and enhanced silver ion release under laser exposure [238]. This composite effectively eliminated mature MRSA biofilms, reducing biomass by approximately 70%—a markedly better outcome compared to bismuth-MSNs alone (27%) or the full construct without laser treatment (31%). In vivo experiments further confirmed the construct’s photothermal-enhanced antibacterial activity, achieving around 96% bacterial elimination in abscess models after a single treatment [238].

Additionally, photodynamic therapy (PDT) represents a promising strategy to combat bacterial resistance, although clinical use has been hampered by limited tissue penetration and concerns over photosensitizer biosafety [239,240]. For example, a nanocarrier system was engineered using tetraethyl orthosilicate and 3-aminopropyltriethoxysilane as precursors to produce PEGylated aminated mesoporous silica nanoparticles, which were further functionalized with hypericin to form a photodynamic antibacterial platform [241]. Upon irradiation, the system exhibited significantly enhanced antibacterial efficacy compared to free hypericin, owing to elevated ROS generation. This construct was effective against both *S. aureus* and *E. coli*, with greater sensitivity observed in *S. aureus*, likely due to differences in cell wall structure [241]. Furthermore, a near-infrared (NIR)-triggered photodynamic system was implemented in an attempt to address drug resistance by utilizing mesoporous silica-coated, lanthanide-doped upconversion nanoparticles (UCNPs) loaded with the photosensitizer erythrosine [141]. This system also effectively generated ROS upon NIR irradiation, disrupting the membranes of *S. aureus* and *E. coli* while demonstrating strong photodynamic antibacterial activity and excellent biosafety [141].

### 3.3. Application in Proteins, Genes and Antigens Delivery

Gene therapy, which aims to treat or cure specific diseases by introducing functional gene segments or regulatory oligonucleotides into a patient’s cells, has become a crucial strategy in targeted therapeutics [242,243,244,245]. To date, the U.S. FDA has approved over 300 biologic drugs, with approximately 80 derived from peptides [242,244]. Key milestones in the development of peptide therapeutics approved by the U.S. FDA, beginning with the discovery of insulin, are summarized in Figure 6 [14,246]. To be successful, gene therapy requires efficient delivery systems to transport genetic material across cellular membranes, with viral vectors remaining the most commonly used due to their high transfection efficiency [188,247]. However, concerns regarding their immunogenicity, production complexity, and cost have driven interest toward safer, non-viral delivery platforms [244,247]. These alternative systems aim to modulate the expression of nucleic acids such as DNA, mRNA, siRNA, antisense DNA, and microRNA (miRNA), with siRNA showing particular promise in silencing specific gene targets [14,246].

Among emerging delivery tools, cell-penetrating peptides (CPPs), short peptides typically comprising 5–30 amino acids, have garnered significant attention due to their ability to traverse tissue and cellular membranes with minimal disruption, with their versatility making them valuable carriers for nucleic acid-based therapies [247]. In this context, smart drug delivery systems incorporating mesoporous silica nanoparticles with their surface modified with cell-penetrating aptamer have demonstrated great potential for the targeted and controlled release of therapeutic agents, including proteins, genes, and siRNA [8,14,246]. They thus have been widely investigated for combination therapies and gene silencing approaches, particularly within cancer nanomedicine [8,14,246]. These nanoparticles were also engineered to interact closely with immune cells, enabling advanced delivery strategies that improve antigen presentation, promote immune cell adhesion, and overcome challenges such as poor cellular uptake [8,14,246]. Furthermore, chemical modifications to the MSN surface have enhanced siRNA encapsulation, protecting these molecules from enzymatic degradation and preventing premature release [8,246,248,249].

Insulin is one of the important landmarks in peptide therapeutics, critical in the conventional management of diabetes. Although effective in controlling blood glucose, this approach is often accompanied by adverse effects such as hypoglycemia, subcutaneous fat atrophy, and delayed absorption, all of which contribute to reduced patient adherence [250]. To overcome these limitations, glucose-responsive insulin delivery systems have emerged as a promising alternative, capable of autonomously regulate the insulin release based on blood glucose levels, potentially reducing toxicity and improving patient compliance compared to conventional therapies [250]. However, to advance beyond subcutaneous clinical applications, oral insulin formulations have been extensively explored, focusing on key factors such as encapsulation efficiency, protection against enzymatic degradation, and intestinal absorption. One promising platform developed in this context involves mesoporous silica nanorods synthesized via a soft-templating method and subsequently loaded with insulin [251]. In vitro release studies indicated that this system holds potential for controlled insulin delivery, as insulin release in simulated gastric fluid was significantly slower than in simulated intestinal fluid, with the release behavior in both environments governed by Fickian diffusion kinetics [251]. A more advanced nanoplatform utilized dendritic mesoporous silica nanoparticles as insulin carriers, combined with succinylated β-lactoglobulin as a protein-based stabilizer [252]. Just like the previous one, this formulation was also integrated into pH-responsive tablets designed to prevent premature insulin release or degradation in the stomach while enhancing intestinal epithelial transport. Using an in vitro insulin-starvation assay and fluorescent imaging, it was shown that this system facilitated effective insulin uptake across intestinal cells [252]. Another sophisticated glucose-responsive mesoporous silica nanosystem engineered for self-regulated insulin delivery was functionalized with carboxyphenylboronic acid and loaded with insulin, then decorated with sodium alginate to serve as a gatekeeping polymer [253]. Under normal glucose conditions, stable interactions between boronic acid and alginate kept the nanopores sealed. However, these interactions were disrupted under elevated glucose levels, triggering controlled insulin release [253]. In diabetic mice, a single administration of this system effectively maintained normoglycemia for up to 12 h and led to improvements in lipid metabolism and organ health, demonstrating both high therapeutic potential and biosafety [253]. Furthermore, a complementary system was developed for the co-delivery of insulin and cyclic adenosine monophosphate (cAMP) using boronic acid-functionalized mesoporous silica nanoparticles [254]. In this design, gluconic acid-modified insulin, labeled with fluorescein isothiocyanate, served dual functions—as a therapeutic agent and as a gatekeeper to retain cAMP within the nanopores. Upon exposure to saccharides, particularly glucose and fructose, the system released both insulin and cAMP, with the inclusion of the latter intended to enhance and sustain the therapeutic effects of insulin [254]. Release profiles were evaluated under various pH conditions, while in vitro studies demonstrated low cytotoxicity, efficient cellular uptake, and high internalization efficiency across four different cell lines, as confirmed by flow cytometry and confocal microscopy [254].

Interactions of silica nanoparticles with lipid membranes have also been extensively explored in the context of antimicrobial and antiviral antigens delivery [255,256]. For example, using LL-37, an antimicrobial peptide, while comparatively investigating three types of silica nanoparticles, including smooth mesoporous, virus-like mesoporous (featuring a spiky surface topology), and nonporous particles, Haffner et al. [32] revealed that surface topology significantly affects membrane binding and disruption, with virus-like nanoparticles exhibiting superior membrane destabilization, an effect amplified upon LL-37 loading [32]. Compared to smooth particles and free LL-37, the peptide-loaded virus-like nanoparticles caused greater membrane perturbation, as illustrated with the schematic representation in Figure 7. 

Neutron reflectometry showed these particles induced transmembrane defects and facilitated LL-37 integration across both bilayer leaflets, while confocal microscopy and live/dead bacterial assays confirmed that this structural advantage led to enhanced bacterial membrane rupture, particularly against *E. coli* [32]. Overall, these results demonstrate that nanoparticle surface topology plays a critical role in modulating interactions with both model membranes and live bacteria, and identify virus-like mesoporous nanoparticles as promising platforms for delivering antimicrobial peptides [32].

Along the same lines, conventional treatments for autoimmune diseases like multiple sclerosis (MS) typically involve broad-spectrum immunosuppressants, which, while effective at dampening immune activity, carry heightened risks of infection and systemic side effects [257,258]. A more precise therapeutic strategy involves the use of self-antigen-loaded mesoporous nanoparticles to induce antigen-specific immune tolerance [257,258]. In a mouse model of MS—experimental autoimmune encephalomyelitis (EAE)—this approach successfully expanded Foxp3^+^ regulatory T cells and significantly reduced central nervous system (CNS) infiltration by both antigen-presenting cells (APCs) and autoreactive CD4^+^ T cells [259]. The addition of reactive oxygen species (ROS)-scavenging cerium oxide nanoparticles further suppressed APC activation and enhanced the tolerogenic effect. Notably, this combined strategy was capable of reversing paralysis even in mice with advanced-stage EAE [259].

Coming to HIV-1 therapy, lipid-coated mesoporous silica nanoparticles were engineered as a biomimetic platform for targeted delivery of the antiretroviral drugs rilpivirine (RPV) and cabotegravir (CAB) [260]. By integrating the ganglioside lipid GM3 into the lipid shell, these nanoparticles selectively bind to CD169^+^ myeloid cells, which are abundant in secondary lymphoid tissues—key HIV-1 reservoirs. The GM3-functionalized coating also dampened inflammatory cytokine release from human macrophages, enhancing biocompatibility [260]. Encapsulation of both RPV and CAB maintained potent anti-HIV-1 activity for up to 90 days at room temperature, underscoring the formulation’s exceptional stability and obviating the need for cold-chain logistics [260].

Central nervous system infections caused by neurotropic viruses such as rabies, Zika, and poliovirus pose significant therapeutic challenges due to the restrictive nature of the blood–brain barrier, which severely limits the penetration of antiviral agents. To address this barrier, mesoporous silica nanoparticles were engineered for targeted CNS drug delivery by functionalizing them with rabies virus glycopeptide (RVG) and loading them with the broad-spectrum antiviral favipiravir (T-705) [261]. In a mouse model infected with vesicular stomatitis virus, this RVG-modified nanoparticle system achieved efficient BBB translocation, leading to a marked reduction in viral replication and brain viral load with minimal cytotoxicity [261]. Notably, treatment with the RVG-nanoparticles resulted in a 77% survival rate at 21 days post-infection, in stark contrast to a 23% survival rate in untreated controls. Additionally, treated mice exhibited significantly lower levels of viral RNA in brain tissues, underscoring the therapeutic potential of this targeted nanoplatform [261].

While siRNA-based therapies have demonstrated significant potential for gene silencing, the emergence of CRISPR-Cas9 gene editing has marked a transformative shift in genetic medicine, enabling precise and permanent modifications at the DNA level, with its application being particularly compelling in the context of cancer therapy [262]. Mesoporous silica nanoparticles have been implemented as promising carriers for delivering CRISPR-Cas9 components—namely, the Cas9 protein and guide RNA (gRNA)—directly to tumor cells [262]. In these cases, MSNs were engineered to effectively encapsulate these components through the functionalization of their surfaces with tumor-targeting ligands, further enhancing their ability to deliver CRISPR-Cas9 complexes selectively to cancerous tissues [19,262,263]. This targeted approach enabled precise editing of oncogenes or tumor suppressor genes, inhibited tumor progression and restored normal cellular function [19,262,263]. For instance, Labauve et al. [264] developed an MSN-based system capable of co-delivering the Cas9 protein and gRNA, achieving efficient gene knockout in human cells. Their platform significantly improved gene editing efficiency while minimizing off-target effects compared to conventional delivery strategies [264]. In another notable example, Zhao et al. [19] engineered a biomimetic MSN platform coated with tumor cell membranes derived from B16-F10 melanoma cells. This system co-delivered carboplatin (CBP) and a CRISPR-Cas9 plasmid targeting PD-L1. The plasmid was intratumorally injected to induce PD-L1 knockout, reactivating the immune response against the tumor [19]. The cancer cell membrane coating provided homologous targeting capabilities, enhancing uptake by tumor cells and ensuring prolonged release and high biocompatibility [19]. In vivo studies demonstrated that this system not only improved antitumor efficacy but also stimulated immune cell proliferation and further enhanced therapeutic outcomes through the modulation of the tumor microenvironment [19]. These developments also highlighted the synergistic potential of combining MSNs with CRISPR-Cas9 for precise, targeted, and immunologically active cancer gene therapies [19,262,263,264,265].

Antigen-presenting cells (APCs) are essential components of the immune system, responsible for initiating and directing adaptive immune responses by presenting antigens to T cells [266]. The successful activation of a targeted immune response depends on the efficient intracellular delivery of specific antigens to APCs—an essential first step in vaccine development [266,267]. In this context, the accumulation of antigens in lymph nodes and subsequent antigen presentation are critical factors influencing the effectiveness of vaccines, particularly in cancer therapy [266,267,268]. However, effective antigen delivery remains a significant challenge due to enzymatic degradation, rapid clearance by the immune system, and the difficulty of traversing complex biological barriers [266,267,268].

Bacterial outer membrane vesicles (OMVs) are nano-sized, spherical structures naturally released from the outer membrane of Gram-negative bacteria [269,270]. They play multifaceted roles in bacterial physiology, pathogenesis, and host-microbe interactions. OMVs have emerged as versatile immunoadjuvant nanocarriers capable of delivering a variety of biomolecules, including virulence factors, toxins, and genetic material, which contribute to bacterial survival, communication, and disease progression. Additionally, OMVs modulate host immune responses by inducing both pro-inflammatory and immunosuppressive effects [269,270], and thus have the capacity to reprogram the tumor microenvironment and activate immune pathways. 

To validate this approach, Lin et al. [271] developed a nanotherapeutic platform that integrates immunostimulatory cytosine–phosphate–guanine oligodeoxynucleotides into MSNs cloaked with OMVs for cancer immunotherapy. This nanohybrid system was designed to enable precise tumor targeting, enhance the production of antitumor cytokines such as IFN-γ, and suppress the immunosuppressive cytokine TGF-β, thereby reshaping the tumor microenvironment. This construct effectively promoted M1 macrophage polarization, dendritic cell maturation, and the development of long-lasting, tumor-specific immune memory—resulting in significant tumor regression with minimal systemic toxicity, and demonstrated therapeutic efficacy against both metastatic and solid tumor models, including 4T1 breast cancer and MC38 colorectal cancer [271]. Transcriptomic analysis revealed that this platform enhanced mitochondrial oxidative phosphorylation in T cells within tumor-draining lymph nodes, alleviating T cell exhaustion and restoring metabolic fitness. These findings underscore the strong potential of this modular nanoplatform to modulate both innate and adaptive immunity in cancer immunotherapy [271].

Additionally, hollow mesoporous silicon nanoparticles loaded with the targeted drug sorafenib were also coated with platelet membranes and anti-PD-1 antibodies to enhance precision drug delivery within the tumor microenvironment and enable effective cancer immunotherapy [272]. These biomimetic nanoparticles retained the immune-evasive properties of natural platelet membranes. Notably, the anti-PD-1-coated nanocarriers demonstrated stronger tumor-targeting affinity than sorafenib-loaded nanoparticles alone [272]. Intravenous administration into hepatocellular carcinoma (HCC) mouse models showed that this system not only directly activated cytotoxic T cells but also promoted the efficient and triggered release of sorafenib [272].

A separate strategy involved the development of a doxorubicin-COOH-loaded mesoporous silica nanoparticle, functionalized with polyethylenimine and nucleic acid chimeras, for the targeted treatment of bladder cancer [273]. This multifunctional nanocarrier was designed to co-deliver siRNA and chemotherapy agents, thereby reducing drug toxicity and overcoming chemoresistance. The chimera-based formulation exhibited controlled doxorubicin release for over 48 h, with significantly lower burst release compared to conventional doxorubicin-loaded mesoporous silica systems [273]. Immunohistochemical analysis confirmed selective binding of the chimera nanocarrier to bladder cancer cells, resulting in notable inhibition of PI3K expression and tumor cell proliferation. This system induced greater apoptosis in doxorubicin-resistant cells than either aptamer-doxorubicin or non-targeted formulations [273]. In vivo studies further demonstrated a marked reduction in tumor volume and increased apoptotic cell ratio, while histological analysis of the kidneys and lungs indicated minimal damage to healthy tissues—highlighting its safety and therapeutic potential [273].

Cancer vaccines are designed primarily to stimulate antitumor adaptive immune responses capable of recognizing and destroying tumor cells. One promising approach involves dendritic cell (DC)-based cancer vaccines, which harness the body ‘s own immune system to combat cancer. However, current DC-based vaccines face several limitations, primarily stemming from the challenges associated with the ex vivo culture and manipulation of patient-derived dendritic cells [274].

To mitigate some of these issues, recent approaches have focused on the direct activation and maturation of host DCs using advanced antigen-delivery platforms. One such strategy involves the use of VMSNs co-loaded with tumor antigens and toll-like receptor 9 (TLR9) agonists. Mimicking viral morphology, these VMSNs are more efficiently internalized by APCs than their non-viral structured counterparts. Upon uptake, they stimulate APC activation via the TLR9 pathway and facilitate cross-presentation of antigens such as ovalbumin (OVA) [186]. In vivo studies demonstrated that VMSN-based nanovaccines significantly enhanced CD4^+^ and CD8^+^ T cell responses in lymph nodes and spleen, while promoting the formation of effector memory T cells, with the schematic illustration of the mechanism of such immune system activation shown in Figure 8 [186]. This led to effective suppression of tumor progression and prolonged survival in B16-OVA tumor-bearing mice under both prophylactic and therapeutic conditions. Furthermore, combining these nanovaccines with immune checkpoint blockade (ICB) therapy resulted in synergistic antitumor effects, highlighting VMSNs as highly promising platforms for cancer immunotherapy [186].

In a related study, another variant of VMSNs was engineered to mimic the radialized spike topology and rough surface of viral particles [185]. These MSNs were co-loaded with the TLR7/8 agonist imiquimod and OVA antigens. Compared to conventional spherical MSNs, these virus-like particles demonstrated superior biocompatibility, enhanced cellular uptake, and an alternative endocytic pathway that facilitated lysosomal escape and promoted antigen cross-presentation, underscoring the critical role of nanoparticle surface topology in generating robust cellular immune responses [185]. In B16-OVA tumor models, this system significantly activated DC maturation, elevated CD8^+^ T cell levels, and suppressed tumor growth [185]. 

Another innovative design involved *silicasomes*, which are mesoporous silica nanoparticles coated with a phospholipid bilayer to improve lymph node targeting and delivery efficiency. Upon subcutaneous injection, these lipid-coated MSNs demonstrated significantly enhanced lymph node accumulation compared to uncoated particles [275]. Their efficacy was validated in B16-OVA tumor models using the OVA257–264 peptide antigen, co-delivered with the TLR4 agonist monophosphoryl lipid A (MPLA) embedded in the lipid membrane. This formulation induced a potent antitumor immune response, characterized by increased infiltration of antigen-specific T cells into tumors [275]. These results demonstrate that lipid-coated MSNs not only improve antigen and adjuvant delivery to lymph nodes but also potentiate immune activation, highlighting their potential as versatile and effective platforms for cancer immunotherapy [275].

To further address the limitations of ex vivo DC manipulation, extra-large pore mesoporous silica nanoparticles developed to directly activate host DCs in situ, offered significantly greater loading capacity for both protein antigens and TLR9 agonists compared to conventional small-pore MSNs [36]. In vitro studies indicated that they enhanced DC activation, antigen presentation, and pro-inflammatory cytokine secretion, while in vivo, these nanocarriers efficiently targeted draining lymph nodes, stimulated antigen-specific cytotoxic T lymphocytes, and suppressed tumor growth following vaccination [36]. Notably, tumor rechallenge experiments in vaccinated mice showed sustained tumor immunity, supported by robust memory T cell responses, confirming that extra-large pore MSNs are a potent and scalable platform for cancer vaccine delivery [36].

Beyond oncology, the virus-like nanoparticle platform has also been explored as a promising alternative to traditional inactivated vaccines. For example, in the context of foot-and-mouth disease, hollow mesoporous silica nanoparticles were employed as nanocarriers for protein-based vaccines [276]. These particles demonstrated excellent biocompatibility, high antigen-loading capacity, and elicited strong antibody responses and protective immunity comparable to commercial adjuvants such as ISA 206 [276]. Furthermore, a universal vaccine delivery platform was developed using virus-like magnetic mesoporous silica nanoparticles combined with synthetic biology-derived, endoplasmic reticulum-targeting vesicles [277]. This system featured a high antigen-loading capacity and enhanced antigen presentation in dendritic cells by directing antigens to the endoplasmic reticulum, thus amplifying immune activation. Following a prime-boost regimen and stimulation with an alternating magnetic field, the nanoplatform elicited robust antibody responses against fungal, bacterial, and viral antigens, conferring significant protection in a systemic fungal infection model, and demonstrating the potential of this magnetism-activated vaccine platform for broad-spectrum immunization [277]. 

Additionally, to enhance mucosal vaccine efficacy, a targeted nano-delivery system was designed to direct antigens to M cells, which specialize in transcytosis in mucosal tissues. This platform employs MSNs coated with a mucoadhesive chitosan-catechol (Chic) layer and conjugated with a recombinant collagenase domain of the porcine epidemic diarrhea virus (PEDV), fused to an M cell-targeting RGD peptide [278]. Produced in *E. coli* and attached to the MSN-Chic surface, this construct demonstrated enhanced mucosal adhesion, efficient M cell targeting, and improved uptake by DCs, while promoting DC trafficking to lymph nodes and increasing CCL20-mediated recruitment of additional DCs [278]. Moreover, this nanocarrier enhanced antigen permeability by modulating the distribution of the tight junction protein ZO-1. Immunization with this system led to strong mucosal and systemic immune responses, including high titers of PEDV-neutralizing antibodies [278]. In a closely related study geared toward addressing the weak immunogenicity observed with traditional vaccines, a genomically active nano-adjuvant (gMSN), composed of β-glucan-modified MSNs loaded with ovalbumin, appeared to facilitate targeted antigen delivery to intestinal lymphoid tissues via M cells upon oral administration, and acted as an epigenetic adjuvant to stimulate potent systemic and mucosal immunity [279]. The platform activated aerobic glycolysis in DCs, enhancing their metabolic fitness and immunostimulatory capacity. It also upregulated key maturation and antigen presentation genes and induced epigenetic remodeling, marked by increased H3K27ac, H3K4me1, and H3K4me3 levels [279]. ATAC-seq revealed increased chromatin accessibility in critical immune-regulatory loci such as *Il2ra*, *Il18r1*, and *Cd83*, suggesting a novel mechanism of immune enhancement through epigenetic reprogramming [279].

To further address critical challenges in oral vaccine delivery—such as antigen degradation in the gastrointestinal (GI) tract, limited penetration of the mucus barrier, and suboptimal adjuvant efficacy—a dendritic fibrous nano-silica (DFNS) platform was engineered by grafting it with *Cistanche deserticola* polysaccharide (CDP). This CDP-functionalized DFNS significantly enhanced antigen uptake and DCs activation in vitro [280]. In vivo, the platform demonstrated efficient accumulation in Peyer ‘s patches, promoted activation of gut-resident immune cells (including DCs, T cells, and B cells), and supported the formation of robust memory T cell responses. Additionally, the system stimulated elevated levels of IgG, IgA, and secretory IgA (SIgA), leading to a well-balanced Th1/Th2 and mucosal immune response [280].

A biomineralization strategy to produce *Bacillus*-shaped mesoporous silica nanoparticles (B-MSs), inspired by natural sponge-like architecture was also implemented in an attempt at creating energy-efficient and scalable vaccine adjuvants [281]. Utilizing *E. coli* engineered to express silicatein as a biological template, B-MSs were synthesized with high surface area and hierarchical porosity, enabling efficient protein loading, with the amino-functionalized variants demonstrating superior protein retention and sustained release, attributed to electrostatic interactions [281]. These amino-functionalized variants significantly enhanced IgG production and CD8^+^ T cell responses—outperforming even complete Freund’s adjuvant in vivo [281].

SARS-CoV-2 variants with immune-evasive mutations have contributed to diminished efficacy of subunit vaccines, largely due to reduced antigen recognition. Toward mitigating this issue, a nanoadjuvant platform was developed using periodic mesoporous organosilica nanoparticles functionalized with trehalose-6,6-dibehenate—a potent Mincle receptor agonist [282]. This system enhanced antigen delivery to lymph nodes and facilitated immune cell uptake. By activating both naïve B cells and dendritic cells via the Mincle signaling pathway, the platform promoted robust germinal center formation and significantly enhanced humoral and cellular immunity [282]. In mice, immunization with this system loaded with the SARS-CoV-2 receptor-binding domain (RBD, wild type) resulted in elevated antigen-specific antibody levels and strong T cell responses [282]. To curtail immune escape by emerging SARS-CoV-2 variants, a complementary vaccine strategy employing dendritic mesoporous silica nanoparticles designed to co-deliver the RBD along with conserved T-cell epitope peptides, was engineered in a related study [283]. This nanocarrier facilitated efficient uptake by antigen-presenting cells and improved lymph node targeting. Immu[284,285nization with this construct induced high titers of RBD-specific neutralizing antibodies and robust cellular responses through effective T-cell activation, highlighting its potential as a broadly protective COVID-19 vaccine candidate [283].

Furthermore, Middle East respiratory syndrome coronavirus (MERS-CoV) remains a high-priority pathogen with a 34% mortality rate and no licensed vaccine, despite extensive global efforts [284,285]. To advance vaccine development, plasmid DNA (pDNA) and mRNA vaccines encoding the MERS-CoV spike protein were formulated using silane-functionalized aluminosilicate nanoparticles for enhanced delivery [286]. In murine models, the codon-optimized, non-encapsulated pDNA vaccine produced the strongest antibody responses, and while nanoparticle encapsulation enhanced the efficacy of the mRNA vaccine, it conferred no additional benefit for the pDNA formulation, suggesting platform-specific considerations in vaccine optimization [286].

In a separate study targeting *Mycoplasma hyopneumoniae*, a respiratory pathogen in pigs, researchers developed an oral vaccine using antigens loaded into mesoporous silica (SBA-15) to complement existing intramuscular vaccines. Sixty piglets were assigned to four groups with varying oral and intramuscular vaccination regimens, and monitored for immune responses, lung lesions, and bacterial shedding up to 73 days of age under high infection pressure [287]. Piglets that received early oral vaccination exhibited the smallest lung lesions, and significantly reduced bacterial shedding by day 61. Interestingly, in orally vaccinated animals, lung pathology did not correlate with bacterial burden, unlike in non-oral groups, positioning SBA-15-based oral vaccination as a practical and effective approach for controlling *M. hyopneumoniae* in swine production [287].

HIV-1 remains a major global health challenge, with no licensed vaccine despite decades of intensive research, although DNA vaccines offer a promising strategy due to their safety, manufacturing simplicity, and low risk of side effects [288,289]. However, their clinical application has been hampered by poor cellular uptake and rapid degradation in vivo [288,289]. To overcome these challenges, CpG-functionalized silica-coated calcium phosphate nanoparticles (SCPs) was introduced as an innovative delivery platform for DNA vaccines encoding native-like trimeric HIV-1 envelope proteins (BG505 SOSIP.664 fused to GCN4 or T4 fibritin motifs) [290]. SCPs enhanced DNA vaccine stability, promoted efficient uptake by antigen-presenting cells, and stimulated robust immune responses. In vivo experiments revealed that SCP-mediated delivery significantly outperformed naked DNA in eliciting both humoral and cellular immunity, generating broader and stronger immune responses [290]. This study represents one of the few successful uses of CpG-functionalized SCPs to deliver trimer-based HIV-1 DNA vaccines, highlighting their potential as a next-generation platform for HIV immunization [290].

While routine vaccination has drastically reduced the incidence of diphtheria and tetanus, conventional aluminum-based adjuvants can trigger adverse effects, prompting the search for safer alternatives. As such, SBA-15 mesoporous silica was investigated as a potential adjuvant in subcutaneous immunization of mice with diphtheria (dANA), tetanus (tANA), and combined diphtheria-tetanus (dtANA) anatoxins. Analytical techniques including SAXS, DLS, NAI, CD/SRCD, and fluorescence spectroscopy confirmed successful encapsulation and preservation of tANA structure within SBA-15 [291]. Immunization trials in both high and low antibody responder mouse strains demonstrated that SBA-15 significantly enhanced primary and booster antibody responses, especially when using combined antigens, with SBA-15 being able to convert low-responder mice into high responders [291].

These representative examples underscore the versatility of MSNs as delivery platforms for a broad range of biological payloads. Their key advantages—including excellent biocompatibility, high loading capacity, protection of sensitive cargo from degradation, controlled release profiles, and ease of surface modification—make them especially promising for advancing protein- and gene-based therapies, as well as for the development of next-generation vaccines. Ongoing research continues to refine and expand these applications, highlighting MSNs ‘ growing relevance in biomedicine.

## 4. Perspective and Challenges

As carefully documented throughout this report via selected examples, MSNs are rapidly emerging as promising platforms in regenerative medicine and in the targeted delivery of therapeutic, diagnostic, and theranostic agents to specific cells and tissues. By tailoring parameters such as particle size, shape, pore dimensions, surface morphology, and post-synthesis modifications, these nanoparticles can be engineered to transport drugs or imaging agents selectively across biological membranes to designated targets [292,293,294]. This versatility has accelerated the growth of research in MSNs for regenerative therapies and drug delivery. In fact, since the first reports of MSN synthesis in 1999 [294,295,296], the field has significantly expanded, with over 10,000 publications exploring MSNs as delivery vectors to date (August 2025). Their tunable properties have enabled precise targeting and synergistic co-delivery of active agents, thereby enhancing therapeutic efficacy. 

One of the major advantages of silica-based nanocarriers is their biocompatibility and tolerability, which help to overcome limitations posed by other delivery systems. However, despite substantial preclinical progress, most MSN-based drug delivery systems have yet to receive regulatory approval and remain in the preclinical stage [297,298]. A few silica-based platforms, nonetheless, have progressed into clinical trials and even gained FDA approval. While many of these latter systems differ from traditional MSNs in porosity and surface architecture, they can still provide important information on the overall safety profile of this family of nanocomposite materials. For instance, AmSil® (amorphous silica nanoparticles) and Cornell dots (C-dots; ultrasmall silica nanoparticles) are typically nonporous or possess alternative pore structures that can significantly influence biodistribution and clearance [294,297,298].

The first clinical application of silica-based nanosystems was reported in 2007 in the treatment of atherosclerotic lesions using plasmonic photothermal therapy (NCT01270139) [8]. This study demonstrated a reduction in target lesion revascularization compared to stent-based treatments and showed no associated cytotoxicity or adverse clinical outcomes [8,294,297]. Cornell dots systems have since also entered Phase I clinical trials for diagnostic imaging applications [299]. Notably, ^124^I-labeled cRGDY silica probes and dye-labeled variants such as cRGDY-PEG-Cy5.5-C have been tested for PET imaging in melanoma, malignant brain tumors (NCT01266096), and nodal metastases (NCT02106598) [297]. Furthermore, C-dots have been evaluated in patients with metastatic melanoma, brain tumors, and head and neck cancers (NCT01266096, NCT02106598, NCT03465618), and some versions such as ^64^Cu-NOTA-PSMA-PEG-Cy5.5-C dots have also entered trials for the identification of tumor cells during prostate cancer surgery (NCT04167969) [297]. Additionally, gold–silica nanoshells developed by Nanospectra Biosciences under the AuroLase platform, which leverage the enhanced permeability and retention effect to accumulate in tumors and enable localized thermal ablation through NIR stimulation following intravenous administration, have also advanced into multiple clinical trials for prostate, head, and neck cancers (NCT00848042, NCT02680535, NCT04240639, NCT04656678) [297]. More importantly, the ultrasmall size of C-dots allows for rapid renal clearance, which mitigates concerns over long-term bioaccumulation [297]. The only reported serious adverse effect, to the best of our knowledge, occurred in a clinical trial involving CD68-targeted microbubbles composed of silica–gold–iron nanoparticles for imaging macrophages in atherosclerotic plaques (NCT01436123), and the study was terminated after patients exhibited signs of toxicity, prompting discontinuation of the therapy [297,298].

Despite this growing body of preclinical research, only a limited number of MSNs have advanced to clinical evaluation—primarily in the context of tumor-targeted therapies [294]. Among these, one notable study investigated the bioavailability of fenofibrate, an antilipidemic agent, when loaded onto MSNs versus the commercially available Lipanthyl® formulation in healthy male volunteers. The results showed favorable tolerance and bioavailability for the MSN-based formulation [300]. Similarly, a silica–lipid hybrid formulation encapsulating simvastatin was assessed for safety, tolerability, and pharmacokinetics in a clinical study involving healthy male participants (ACTRN12618001929291) [301]. The formulation demonstrated a 3.5-fold increase in bioavailability compared to the standard commercial product, with no adverse effects reported [301]. In another trial, ibuprofen-loaded silica–lipid hybrid nanoparticles were administered orally to 16 healthy individuals in a Phase I study. The results indicated a 1.95-fold enhancement in ibuprofen bioavailability, with only mild side effects observed [301].

Nevertheless, while the number of silica-based nanosystems evaluated in human clinical trials remains small, the data to date suggest that silica nanoparticles are generally safe and can enhance the pharmacokinetics and therapeutic efficacy of drug molecules [302], thus supporting the potential of MSNs in future clinical applications. However, FDA approvals have so far been limited to oral delivery systems, with a focus on evaluating biocompatibility and systemic behavior rather than fully validating therapeutic efficacy in complex disease models [302]. Therefore, more extensive clinical trials are essential to bridge the gap between promising preclinical outcomes and real-world clinical translation. Furthermore, despite their classification as “generally recognized as safe” by the U.S. FDA, concerns about potential toxicity—particularly from intravenously administered MSNs—persist. A recent study investigated the role of nanoparticle shape and shear stress in MSN-induced toxicity using an in vitro blood flow model and HUVECs [303]. The results revealed that under physiological flow conditions, rod-like MSNs (R-MSNs) induced mechanical damage to the cell membrane due to shear stress, while spherical MSNs (S-MSNs) caused both mechanical and oxidative stress [303]. In vivo models further supported these findings, showing that both R-MSNs and S-MSNs induced cardiovascular toxicity in zebrafish and mice due to high shear stress, particularly in cardiac tissue. S-MSNs were also associated with oxidative damage at accumulation sites such as the liver, spleen, and lungs, whereas R-MSNs did not induce significant oxidative stress [303]. These findings underscore the importance of nanoparticle shape and dynamic physiological forces in determining the biosafety profile for MSNs.

Biodegradability is another critical consideration influencing the clinical applicability of MSNs. Recent research has highlighted that the biodegradation rate is highly dependent on various physicochemical properties, including surface area, morphology, pore size, particle diameter, and the condensation degree of the silica framework [304,305,306,307]. Typically, conventional MSNs require several days to degrade, with complete excretion taking weeks, depending on the nanoparticle design [304,305,306,307]. Poor biodegradability can lead to bioaccumulation and potential long-term toxicity [304,305,306,307,308], emphasizing the urgent need for improved degradation profiles to support safe in vivo applications. It is worth noting that progress has been made in recent years to enhance MSN biodegradability, including modifying the silica network and incorporating structural defects to tune degradation rates [304,305,306,307,308]. Structural defects—such as vacancies and substitutional impurities—not only endow MSNs with useful physicochemical properties (e.g., optical activity, redox behavior) but also influence their safety and biodegradability. As a result, controlled “defect engineering” has been employed to enhance features relevant for biomedical use, including triggered biodegradability for controlled drug release, catalytic behavior for therapeutic chemical reactions in vivo, paramagnetic properties for MRI, and luminescence for imaging applications [309].

In conclusion, while significant strides have been made toward the clinical use of tailored MSNs in regenerative medicine and theranostics, several key challenges remain. These include optimizing safety profiles, addressing particle shape- and stress-related toxicity, enhancing biodegradability, and ensuring efficient systemic clearance. Nonetheless, the progression of silica-based systems through clinical trials and recent FDA approvals highlight a growing recognition of their potential. It is increasingly likely that MSN-based platforms will see broader clinical adoption in the near future, driven by continued innovation in design, safety validation, and therapeutic integration.

## Figures and Tables

**Figure 1 ijms-26-07918-f001:**
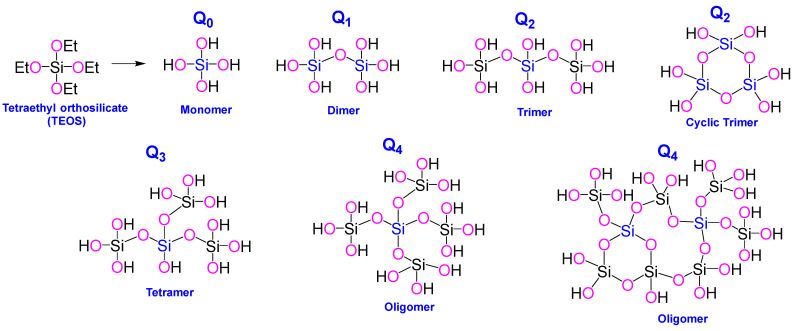
Silicic acid species derived from tetraethyl orthosilicate (TEOS) in solution, ranging from monomeric to oligomeric forms. These species are often described using the Q_n_ notation, commonly employed in ^29^Si-NMR spectroscopy [4,20], where *n* represents the number of adjacent silicon atoms connected through siloxane (Si-O-Si) linkages to a central silicon atom (highlighted in blue). The general empirical formula for these oligomeric structures is Si(OSi)_n_(OH)_4−n_, with *n* values from 0 to 4, corresponding to the Q_0_ through Q_4_ species.

**Figure 2 ijms-26-07918-f002:**
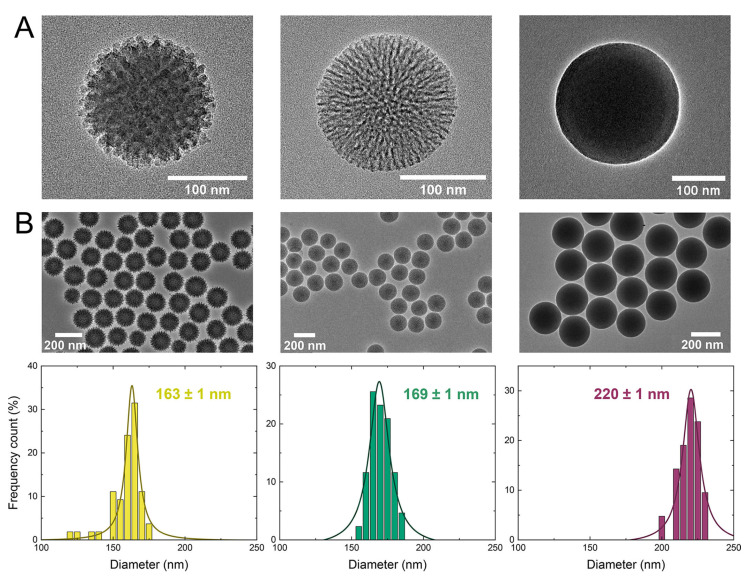
Panel (**A**) presents high-resolution TEM images of three nanoparticle types: virus-like mesoporous (**left**), smooth mesoporous (**center**), and smooth nonporous (**right**) [32]. Panel (**B**) displays low-magnification TEM images capturing a broader population of nanoparticles for each category. The histograms at the bottom summarize the particle size distributions, with Lorentzian fits applied to extract the average particle sizes and associated standard errors for each nanoparticle type (*reproduced with the permission of ACS Nano **2021**, 15, 6787−6800* [32]*, under CC-BY 4.0 license*).

**Figure 3 ijms-26-07918-f003:**
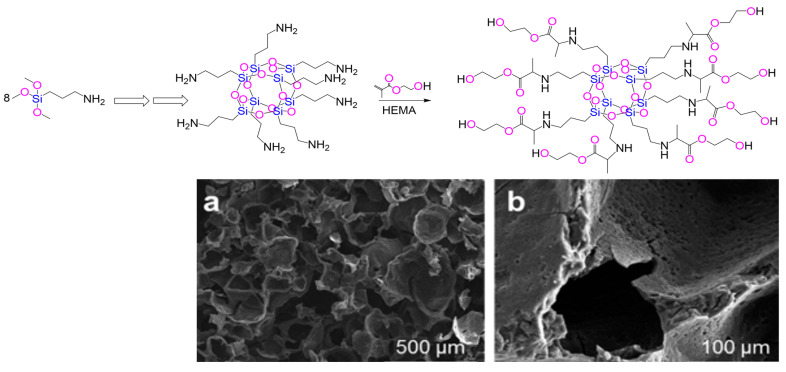
Schematic synthesis and SEM images of 2-hydroxyethyl methacrylate-functionalized self-assembled organic–inorganic biohybrids (500 µm (**a**) and 100 µm (**b**) resolution) for a potential application in bone regeneration (*SEM images reproduced with the permission of New J. Chem., **2018**, 42, 39–47* [25]*, under CC-BY 3.0 license*).

**Figure 4 ijms-26-07918-f004:**
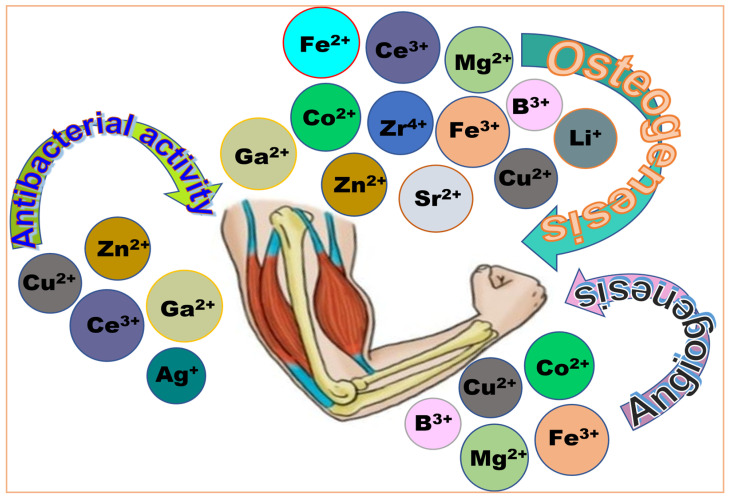
Overview of commonly used ions to support therapeutic and biocidal functions, sanitation, and vascularization in hard- and soft-tissue regeneration [56,73,74].

**Figure 5 ijms-26-07918-f005:**
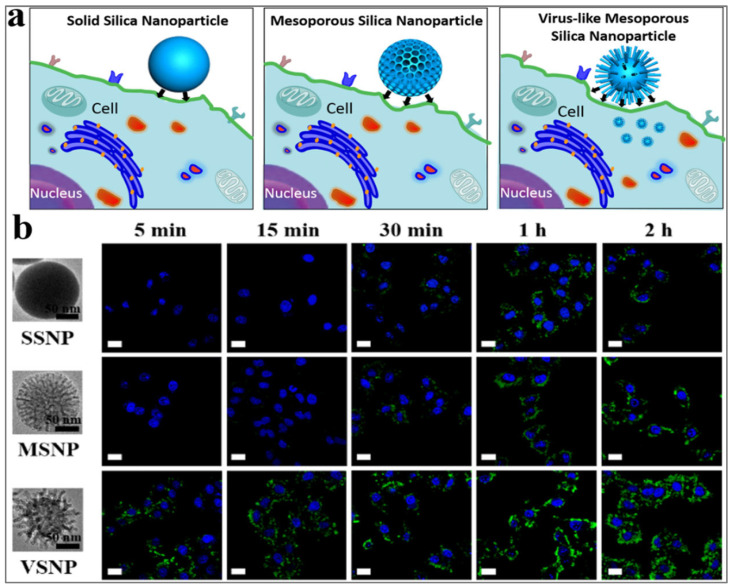
(**a**) Cartoon-style comparison of the cellular internalization for smooth nonporous, smooth mesoporous, and virus-like mesoporous nanoparticles, (**b**) confocal laser scanning microscopy images of HeLa cells following nanoparticles uptake after incubation (*reproduced with the permission of ACS Cent. Sci. **2017**, 3, 839−846* [7]*. Copyright 2017 American Chemical Society. Further permissions related to this material should be directed to the ACS*).

**Figure 6 ijms-26-07918-f006:**
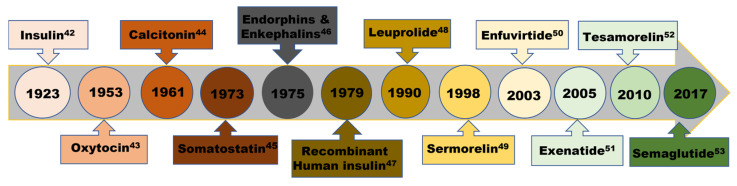
Important milestones in the development and U.S. FDA approval of peptide therapeutics [14,246].

**Figure 7 ijms-26-07918-f007:**
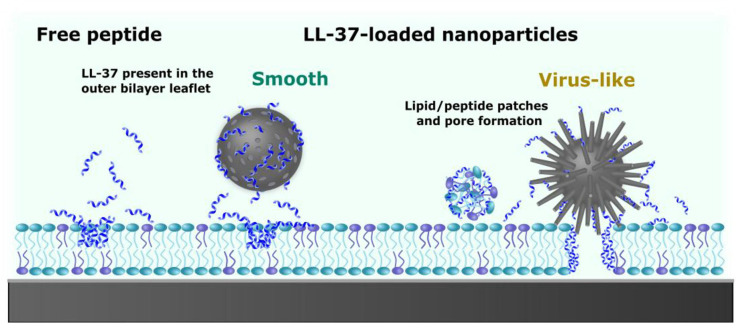
Schematic illustration of bilayer structural alterations detected by neutron reflectivity following exposure to LL-37, either in free form or loaded in smooth or virus-like mesoporous silica nanoparticles, respectively (*reproduced with the permission of ACS Nano **2021**, 15, 6787−6800* [32]*, under CC-BY 4.0 license*).

**Figure 8 ijms-26-07918-f008:**
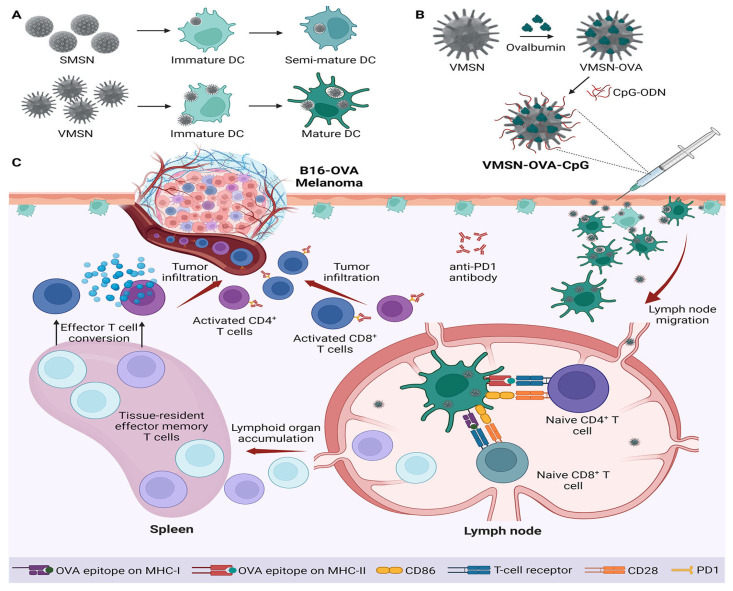
(**A**) Cartoon illustration of a better cellular internalization for virus-like mesoporous silica compared to spherical mesoporous silica; (**B**) Surface decoration of VMNS with ovalbumin; (**C**) Schematic illustration of a virus-mimicking nanovaccine inducing an antigen-specific adaptive immune response against cancer: Following subcutaneous injection, the VMSN nanovaccines migrate to the draining lymph nodes, where they are taken up by tissue-resident immature DCs. This uptake promotes DC maturation, marked by elevated CD86 expression and the presentation of tumor antigens on MHC class I and II molecules. Mature DCs then activate naïve T cells—priming CD8^+^ cytotoxic T lymphocytes via MHC-I and CD4^+^ helper T cells via MHC-II. This leads to the generation of antigen-specific effector memory T cells capable of targeting tumor cells [186]. The T-cell response elicited by the nanovaccine synergizes with anti-PD-1 immunotherapy, effectively suppressing melanoma tumor growth (*reproduced with the permission of ACS Appl. Mater. Interfaces **2024**, 16, 45917−45928* [186]*. Copyright 2024 American Chemical Society. Further permissions related to this material excerpted should be directed to the ACS*).

## Data Availability

As this is a review article, no new data were generated.

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
