# Peer review of "Applications of Tailored Mesoporous Silicate Nanomaterials in Regenerative Medicine and Theranostics"

_ijms, 2025, doi:10.3390/ijms26167918_

Round 1
Reviewer 1 Report
Comments and Suggestions for Authors
- The paragraph "Photodynamic therapy (PDT) represents a promising strategy to combat bacterial resistance; however, clinical use has been hampered by limited tissue penetration and concerns over photosensitizer biosafety..." should be incorporated into the discussion on theranostics applications.
- This manuscript lists numerous examples but lacks in-depth discussion. For instance, regarding this example: "three types of dendritic mesoporous silica nanoparticles with similar particle sizes but varying pore sizes were engineered for degradation in simulated intestinal fluid and used to encapsulate Fenofibrate, a lipid-lowering medication," what are the differences in the loading efficiency and release rate of Fenofibrate among the three types of mesoporous silica nanoparticles?
- In the section on theranostics applications, it is necessary to add content on the use of mesoporous silicate nanomaterials in antibacterial applications.
- For descriptions pertaining to the same reference, it is unnecessary to cite it in every sentence. It is suggested to cite only once per paragraph, such as Ref. 159, Ref. 172, Ref. 193....
- These references should be cited in appropriate locations:Acta Biomaterialia, 2025, doi:10.1016/j.actbio.2025.07.025;Carbon Energy, 2025, doi: 10.1002/cey2.70078;
Author Response
Comment 1: The paragraph "Photodynamic therapy (PDT) represents a promising strategy to combat bacterial resistance; however, clinical use has been hampered by limited tissue penetration and concerns over photosensitizer biosafety..." should be incorporated into the discussion on theranostics applications.
Response: We appreciate this constructive suggestion, which has been implemented through a reorganization of the manuscript. In the process additional examples of smart delivery of antibacterial were added, despite the nature of the load not been a critical parameter for the selection of illustrative examples.
Comment 2: This manuscript lists numerous examples but lacks in-depth discussion. For instance, regarding this example: "three types of dendritic mesoporous silica nanoparticles with similar particle sizes but varying pore sizes were engineered for degradation in simulated intestinal fluid and used to encapsulate Fenofibrate, a lipid-lowering medication," what are the differences in the loading efficiency and release rate of Fenofibrate among the three types of mesoporous silica nanoparticles?
Response: This is a valid and appreciated point. However, given the breadth of this research area, determining the appropriate level of depth is challenging. The manuscript already spans 52 pages, focusing on selected examples and the degree of detail we deemed appropriate for each approach. Including a detailed discussion of loading efficiency for one construct would necessitate doing the same for all others to maintain consistency. These specific data points are readily accessible to interested readers from the cited original reference if needed. In the particular case cited by reviewer, our focus was on illustrating the impact of particle size on degradability, plasma lifetime, and body excretion. We understand that the reviewer’s example is meant to emphasize the importance of more details, but incorporating such depth across all constructs would compromise the readability and fluidity of the report. We believe the current balance achieves both clarity and engagement without overwhelming the reader.
Comment 3: In the section on theranostics applications, it is necessary to add content on the use of mesoporous silicate nanomaterials in antibacterial applications.
Response: This is another valid point, and additional examples of smart antibacterial delivery have been included. It is important to note, however, that the nature of the therapeutic payload is not the primary criterion for selecting illustrative examples; rather, the emphasis is placed on the ingenuity of the design.
Comment 4: For descriptions pertaining to the same reference, it is unnecessary to cite it in every sentence. It is suggested to cite only once per paragraph, such as Ref. 159, Ref. 172, Ref. 193....
Response: While the reviewer is absolutely correct, the decision on how often to insert a given reference depends largely on the readership. At times, it may be unclear to some readers whether a statement within a paragraph reflects the author’s opinion or is drawn directly from a source. In such cases, when I sense potential ambiguity, I find it helpful to include a reference for clarity. That said, I am not aware of a strict rule governing this practice, and it often comes down to editorial judgment.
Comment 5: These references should be cited in appropriate locations: Acta Biomaterialia, 2025, doi:10.1016/j.actbio.2025.07.025;Carbon Energy, 2025, doi: 10.1002/cey2.70078.
Response: While one of these references addresses the degradation behavior of nanomaterials, the other appears to fall completely outside the scope of this review. In either case, it's unclear whether they would add any meaningful value to the report.
Reviewer 2 Report
Comments and Suggestions for Authors
Recent advancements in surface-modified nanosystems for bioengineering and biomedical applications highlight silica-based nanomaterials as promising candidates, thanks to their biocompatibility, biodegradability, and ease of functionalization. This review summarizes cutting-edge developments and applications of these materials in regenerative medicine and theranostics, showcasing their versatility with specific examples. Overall, this review is comprehensive and recommended for publication after revision. Herein, there are several comments for the revision.
- In this review, the perspectives and challenges have been provided. The author should reveal the information at the end of the introduction. Including these points will provide a clearer direction for future research and emphasize the significance of the discussed topics.
- For the section “2. Application of mesoporous silicate in regenerative medicine”, the recent applications should be summarized in a table. This table can assist readers in understanding the advancements.
- Similarly, for the section “3. Application of mesoporous silicates in theranostics”, the recent applications should be summarized in a table.
- For the introduction “These characteristics allow for fine-tuning of pore sizes, making it highly adaptable for various delivery applications”, more references can be cited to highlight its importance.
https://doi.org/10.1021/acsnano.5c02541
https://doi.org/10.1186/s12951-023-02208-3
Author Response
Comment 1: In this review, the perspectives and challenges have been provided. The author should reveal the information at the end of the introduction. Including these points will provide a clearer direction for future research and emphasize the significance of the discussed topics.
Response: The manuscript was revised accordingly through gentle modifications to the abstract.
Comment 2: For the section “2. Application of mesoporous silicate in regenerative medicine”, the recent applications should be summarized in a table. This table can assist readers in understanding the advancements.
Response: In many reviews, the pursuit of exhaustiveness often leads to the inclusion of tables, under the assumption that they summarize information, provide clearer direction, and emphasize the significance of the topics discussed. However, such tables often end up as mere lists of approaches and references, lacking depth and clarity. This review does not aim to be exhaustive; instead, it seeks to offer a balanced perspective on recent applications of bioengineered silica-based nanomaterials in regenerative medicine and theranostics, illustrated through carefully selected examples placed within their appropriate contexts. In this light, the inclusion of tables would not enhance the scientific soundness or overall impact of this report. We are however pleased that the reviewer characterizes this report as comprehensive.
Comment 3: Similarly, for the section “3. Application of mesoporous silicates in theranostics”, the recent applications should be summarized in a table.
Response: Same comments as above.
Comment 4: For the introduction “These characteristics allow for fine-tuning of pore sizes, making it highly adaptable for various delivery applications”, more references can be cited to highlight its importance.
https://doi.org/10.1021/acsnano.5c02541
https://doi.org/10.1186/s12951-023-02208-3
Response: The reviewer is correct that additional references can always be included. However, the central question is whether their inclusion would meaningfully enhance the scientific value, clarity, or overall soundness of the report. In this case, it is not evident that doing so would provide such benefits.
Reviewer 3 Report
Comments and Suggestions for Authors
The article "Applications of Tailored Mesoporous Silicate Nanomaterials in Regenerative Medicine and Theranostics" ambitiously aims to describe the current knowledge in the domain of mesoporous structures based on various silicates and silica structures with medical applications. It is a valuable study that can be published after authors address the following problems:
While the objective of this review is clear, authors should indicate also the review methodology (keywords used, databases consulted, years interval considered, other criteria). For review criteria please see PRISMA for example.
All Latin names should be italicized across the manuscript.
Please add bio-glass as keyword to improve the hit chances in search engines.
Under section 2.2 please discuss more and update with doi: 10.3390/pharmaceutics16091225 on features like biocompatibility and antimicrobial activity.
Please pay attention to the nomenclature, especially silica vs silicate. Indication of the cations should be made where the corresponding silicates are discussed (e.g calcium silicate, zinc silicate, sodium silicate, aluminosilicate etc). Were organic functionalized (modified) silica is the subject, please refrain to name it silicate. Please read https://goldbook.iupac.org/terms/view/OT07579.
Once an abbreviation explicated is not necessary to do it again (e.g. ROS at rows 431 and 456). Other abbreviations are not explicated (e.g MRI).
I also suggest that the authors familiarize themselves with the work of Petrisor G et al. on various mesoporous silica types loaded with polyphenols or plant extract to increase the bioavailability.
References 77 and 78 have same title, same authors and same journal, at 10 years apart. The paper from 2023 is a correction of the 2013 paper and is indicated at the beginning of the 2013 paper. Please check the journal citation rules if is mandatory to cite both article + correction.
Author Response
Comment 1:
The article "Applications of Tailored Mesoporous Silicate Nanomaterials in Regenerative Medicine and Theranostics" ambitiously aims to describe the current knowledge in the domain of mesoporous structures based on various silicates and silica structures with medical applications. It is a valuable study that can be published after authors address the following problems:
While the objective of this review is clear, authors should indicate also the review methodology (keywords used, databases consulted, years interval considered, other criteria). For review criteria please see PRISMA for example.
Response: This review primarily explores how endogenous or exogenous triggers can be leveraged to achieve selective and precise delivery of various biomolecules and active therapeutics across diverse cellular environments, by harnessing the intrinsic properties of mesoporous silicate nanoparticles. This focus also guided the selection of the examples discussed throughout the report, and accordingly, keywords were adapted for each paragraph, and no time constraints were imposed. This has been clarified in the abstract.
Comment 2: All Latin names should be italicized across the manuscript.
Response: It was once standard practice to italicize all Latin words in a manuscript. However, many of these terms have become so commonly used in English that italicization is no longer necessary. I will defer to the editorial judgment on this matter.
Comment 3: Please add bio-glass as keyword to improve the hit chances in search engines.
Response: Done
Comment 4: Under section 2.2 please discuss more and update with doi: 10.3390/pharmaceutics16091225 on features like biocompatibility and antimicrobial activity.
Response: Additional examples of smart delivery of antibacterial were added, despite the nature of the load not been a critical parameter for the selection of illustrative examples.
Comment 5: Please pay attention to the nomenclature, especially silica vs silicate. Indication of the cations should be made where the corresponding silicates are discussed (e.g calcium silicate, zinc silicate, sodium silicate, aluminosilicate etc). Were organic functionalized (modified) silica is the subject, please refrain to name it silicate. Please read https://goldbook.iupac.org/terms/view/OT07579.
Response: The term silicate has a much broader scope than silica, and while modified silica materials can often be classified as silicates, the reverse is not always true. Although I concur with the reviewer’s point, a great care was taken while writing manuscript to distinguish between the two terms, and I believe the current wording across the manuscript is appropriate.
Comment 6: Once an abbreviation explicated is not necessary to do it again (e.g. ROS at rows 431 and 456). Other abbreviations are not explicated (e.g MRI).
Response: The manuscript has been revised in light of this comment.
Comment 7: I also suggest that the authors familiarize themselves with the work of Petrisor G et al. on various mesoporous silica types loaded with polyphenols or plant extract to increase the bioavailability.
Response: While the work by Petrisor G. et al. is commendable, its inclusion would not provide significant scientific value to this report.
Comment 8: References 77 and 78 have same title, same authors and same journal, at 10 years apart. The paper from 2023 is a correction of the 2013 paper and is indicated at the beginning of the 2013 paper. Please check the journal citation rules if is mandatory to cite both article + correction.
Response: The 2023 reference builds upon the 2013 work, not necessarily as a correction, but as a continuation and advancement of the study in light of recent scientific developments. It is therefore appropriate to cite both references together.